# Bioactive-Glass-Based Materials with Possible Application in Diabetic Wound Healing: A Systematic Review

**DOI:** 10.3390/ijms25021152

**Published:** 2024-01-17

**Authors:** Marian Vargas Guerrero, Floor M. A. Aendekerk, Candice de Boer, Jan Geurts, Jimmy Lucchesi, Jacobus J. C. Arts

**Affiliations:** 1Department of Orthopaedic Surgery, Maastricht University Medical Centre (MUMC+), 6229 HX Maastricht, The Netherlands; marian.vargasguerrero@maastrichtuniversity.nl (M.V.G.); f.aendekerk@student.maastrichtuniversity.nl (F.M.A.A.); candice.deboer@maastrichtuniversity.nl (C.d.B.); j.geurts@mumc.nl (J.G.); 2Laboratory for Experimental Orthopaedics, Faculty of Health, Medicine & Life Sciences, Maastricht University, 6229 ER Maastricht, The Netherlands; 3Bonalive Biomaterials, 20750 Turku, Finland; jimmy.lucchesi@bonalive.com; 4Department of Orthopaedic Biomechanics, Faculty of Biomedical Engineering, Eindhoven University of Technology, 5612 AZ Eindhoven, The Netherlands

**Keywords:** diabetic wounds, bioactive glass, wound healing, angiogenesis, osteomyelitis

## Abstract

Diabetes affected 537 million adults in 2021, costing a total of USD 966 billion dollars in healthcare. One of the most common complications associated with diabetes corresponds to the development of diabetic foot ulcers (DFUs). DFUs affect around 15% of diabetic patients; these ulcers have impaired healing due to neuropathy, arterial disease, infection, and aberrant extracellular matrix (ECM) degradation, among other factors. The bioactive-glass-based materials discussed in this systematic review show promising results in accelerating diabetic wound healing. It can be concluded that the addition of BG is extremely valuable with regard to the wound healing rate and wound healing quality, since BG activates fibroblasts, enhances M1-to-M2 phenotype switching, induces angiogenesis, and initiates the formation of granulation tissue and re-epithelization of the wound. In addition, a higher density and deposition and better organization of collagen type III are seen. This systematic review was made using the PRISMA guideline and intends to contribute to the advancement of diabetic wound healing therapeutic strategies development by providing an overview of the materials currently being developed and their effect in diabetic wound healing in vitro and in vivo.

## 1. Introduction

Diabetes is a chronic metabolic disorder that affects a significant number of people worldwide. In 2021 alone, there were 537 million adults living with diabetes, and this number is expected to rise to 643 million by 2030 and 783 million by 2045. The cost of diabetic healthcare in 2021 was at least USD 966 billion dollars, corresponding to an increase of 316% compared to the last 15 years [1]. Diabetes can be classified into type 1 and type 2: diabetes type 1 is caused by a lack of insulin secretion by beta cells of the pancreas, while diabetes type 2 is caused by a deceased sensitivity of target tissues to insulin [2]. Type 2 diabetes is the most prevalent, accounting for over 90% of all diabetes cases globally. Hyperglycemia is a hallmark of type 2 diabetes, but its exact onset time is often challenging to determine, leading to a substantial number of undiagnosed individuals within the population. Delayed diagnosis can result in severe complications, including visual impairment, poorly healing lower-limb ulcers, heart disease, or stroke [1,3,4].

One of the notable complications associated with diabetes is the development of chronic skin ulcers, particularly on the lower extremities, referred to as diabetic foot ulcers (DFUs). These ulcers form due to a combination of factors, such as a lack of feeling in the foot, poor circulation, foot deformities, irritation (such as friction or pressure), and trauma, as well as the duration of diabetes [5]. DFUs affect around 15% of diabetic patients and, if left untreated, can lead to lower extremity amputation (14–24%), with a mortality rate of 50–59% within five years post-amputation. Patients with DFU can experience discomfort, swelling, drainage, and skin changes in the surrounding areas [6], and the foot ulcer can also cause a loss of mobility, which can lead to depression [7]. The impaired healing of diabetic ulcers can be attributed to a range of factors, including pre-existing conditions such as peripheral neuropathy, arterial disease, immunodeficiency, infection, and aberrant extracellular matrix (ECM) degradation [8,9,10]. 

To find solutions that help with diabetic wound healing, it is important to understand how the biological healing process works. Wound healing refers to the process of restoring the integrity and function of the skin and underlying tissues after a break in the epithelial layer [11]. It consists of four overlapping stages, those being hemostasis, inflammation, proliferation, and remodeling (Figure 1). These stages are described by Leaper and Enoch, (2008) as follows [11]: I.**Hemostasis**: This occurs immediately after tissue injury. The coagulation cascade initiates, leading to clot formation and platelet aggregation.II.**Inflammation:** This is divided into an early (day 1–2) and late phase (day 2–3), with neutrophils arriving at the wound site during the early phase to contain microorganisms and initiate repair by activating local fibroblasts and epithelial cells. During the late inflammatory phase (days 2–3), macrophages produce growth factors, facilitating fibroblast proliferation, smooth muscle cells, and endothelial cells, inducing angiogenesis [12].III.**Proliferation:** This stage starts around day 3 and lasts from 2 to 4 weeks after wounding. It is characterized by fibroblast migration, deposition of the extracellular matrix (ECM), and formation of the granulation tissue. Epithelialization of the wound represents the final stage of the proliferative phase.IV.**Remodeling:** During this stage, there is a continuous synthesis and breakdown of collagen as the extracellular matrix is constantly remodeled. This comes to an equilibrium about 21 days after wounding. Collagen type III is replaced by collagen type I.
Figure 1Wound healing stages. (**a**) Processes that occur during the different wound healing stages. I. Hemostasis: occurs immediately after tissue damage to stop bleeding by platelet aggregation and the coagulation cascade. II. Inflammation: removal of the bacteria and dead cells in wound bed by inflammatory cells (neutrophils and macrophages). Triggers the proliferation stage. III. Proliferation: proliferation and migration of fibroblasts, keratinocytes, and endothelial cells. Re-epithelialization, angiogenesis, and granulation tissue formation. IV. Remodeling: reorganization of ECM, where collagen type III is replaced by collagen type I. This can last one year or more [13]. ECM: extracellular matrix; MP: metalloproteinases; TIMP: tissue inhibitors of metalloproteinases [11]. (**b**) Four phases of wound healing. Each stage is key in order to ensure a proper wound healing response [14].
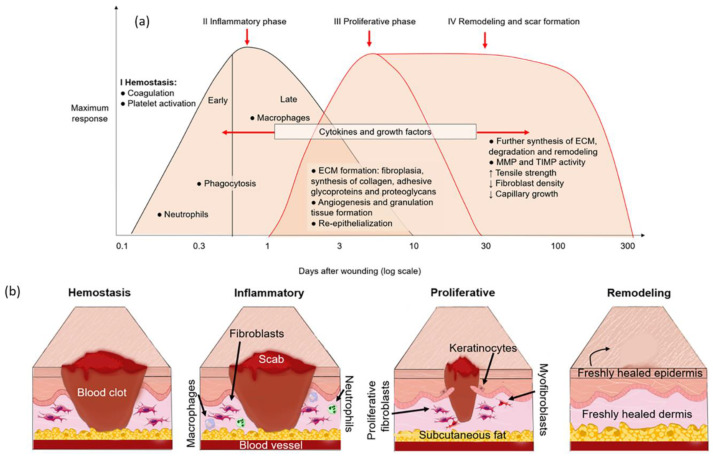


Non-healing wounds can be due to persistent inflammatory response, poor angiogenesis and tissue regeneration, excessive wound exudate, repeated or persistent bacterial infection, and an accumulation of excess reactive oxygen species (ROS) [15]. Surgical debridement, local/systemic application of anti-inflammatory drugs, or antibiotics and wound dressings are widely used in the clinical treatment of these type of wounds. These treatments can alleviate the deterioration of the disease and reduce the patients’ pain to a certain degree, but the overall cure rate is less than 50% [16]. The long-term use of antibiotics may lead to pathogens developing drug resistance [17]. Traditional wound dressings such as gauze, hydrocolloids, films, and foams are single-function, meaning that they act as a physical barrier or absorb exudates but cannot meet all the needs of the wound healing process [16]. Diverse methods, including larval therapy and revascularization approaches, have been employed to enhance the healing process of ischemic ulcers. These methods have exhibited efficacy in mitigating pain, augmenting arterial blood flow to the ischemic extremity, and diminishing the likelihood of amputation [18]. However, there is currently a lack of dressings that can promote angiogenesis in the wound site while at the same time having antimicrobial properties to prevent infection [19]. 

To address the challenges in diabetic soft tissue wound healing, researchers have explored the potential of bioactive-glass-based materials. A bioactive material is defined as a material that elicits a specific biological response at the interface of the material, which results in the formation of a bond between the tissue and the material [20]. Bioactive glass stimulates a specific biological response at the interface with host tissues and has shown promise in bone regeneration [21,22,23]. For soft tissue, it has been shown that specific bioactive glass compositions can facilitate wound healing by promoting angiogenesis, having antibacterial properties [24] and an ability to enhance the M1-to-M2 macrophage switch [25], accelerating the wound healing process. It is important to mention that not all bioactive glasses are the same; they have different compositions and structures, and therefore different properties. Glass has network formers: these components are able to form the glass without the need for additional components and bioactive glass, and they include silica (SiO_2_), phosphorus pentoxide (P_2_O_5_), and boron trioxide (B_2_O_3_). There are also network modifiers, which are not fundamental for the glass structure: they modify the glass structure, allowing the material to have different properties, and they are generally oxides of alkali or alkaline earth metals. The presence of these network modifiers opens the silicate network by breaking Si-O-Si, which causes the pronounced differences in properties of bioactive glass and conventional glass [26,27]. Because of the properties previously mentioned, bioactive glasses are increasingly being investigated for their application in soft tissue regeneration. These bioactive glass materials release ions that can stimulate cell functions, induce angiogenesis [28,29], and exhibit antimicrobial properties [30,31], making them suitable for different stages of the wound healing process [13]. 

It is still unclear in which wound healing stage bioactive-glass-based materials have the most impact on diabetic wound healing. The systematic study of the currently available materials can lead to a further understanding of the effects of bioactive-glass-based materials in diabetic wound healing, allowing us to advance in the field.

Given the increasing interest in utilizing bioactive glass in soft tissue regeneration, this review aims to create an overview of the composition and purpose of the novel bioactive-glass-based materials developed for the treatment of diabetic soft tissue wounds. In addition, the cytocompatibility and angiogenic and antimicrobial properties are evaluated in vitro and in vivo. The diabetic wound closure and wound healing quality of the different bioactive-glass-based materials is evaluated in vivo. This systematic review intends to contribute to the advancement of diabetic wound healing therapeutic strategy development. 

## 2. Methods

The Preferred Reporting Items for Systematic Reviews and Meta-Analyses (PRISMA) statement 2020 for reporting systematic reviews and meta-analyses of studies that evaluate health care interventions has been followed for this systematic review [32]. The systematic review protocol was submitted to the Open Science Framework (OSF), and it can be found through the following link: https://osf.io/bfhpm (accessed on 9 March 2023).

### 2.1. Eligibility Criteria 

In vitro and in vivo studies that used or fabricated novel bioactive-glass-based biomaterials to aid diabetic wound healing were considered for inclusion. Systematic reviews, meta-analysis, literature studies, gray literature, non-peer reviewed journals, and case reports were excluded. Studies were eligible for inclusion when an English full text was available and targeted the research question. Studies were grouped based on the dressing for the syntheses, those being hydrogel, electrospun fibers, and miscellaneous, which included scaffolds, ointments, bioactive glass extracts, and composites.

### 2.2. Information Sources and Search Strategies 

An advanced literature search was carried out using PubMed, EMBASE, Web of Science, and Scopus to find relevant papers on the synthesis and use of novel bioactive-glass-based biomaterials to aid diabetic wound healing. In general, the search strategy included specific key words concerning the bioactive-glass-based materials, diabetes, and diabetic wound. The literature searches were performed independently by two reviewers, MV and FA, and the final search was determined and executed on 27 May 2023. The full search strategies and MESH terms are described in Appendix A. 

### 2.3. Selection Process

After the literature search was performed for all databases, the studies were examined for eligibility. First, all studies were uploaded on RAYYAN and duplicates were removed. Subsequently, the remaining studies were screened based on title and abstract using RAYYAN and included or excluded for full-text screening based on the set eligibility criteria set by two researchers, MV and FA. The screening was performed independently by the two reviewers and compared afterwards. After the screening, the full text of the included studies was assessed independently for final inclusion by the two reviewers, MV and FA. In case of inconsistencies, a third reviewer was consulted for consensus (CA). The selection process was reported in a PRISMA flow chart. 

### 2.4. Data Collection Process and Data Items 

To extract data from the included studies, a standardized data extraction form was made in an Excel spreadsheet by MV and FA. MV collected the data of the first half and FA collected the data of the second half of the included studies independently. To confirm that data were correctly extracted, two papers of the other reviewer’s studies were randomly chosen for data extraction and compared to the work of the other researcher, and there was 100% agreement in the data extracted by both reviewers. 

The general data items that were collected for data extraction included DOI, article title, year of publication, country, statistical analysis, funding received, and possible conflicts of interest. Data items concerning the bioactive glass (BG) properties included composition, function of BG in the dressing, particle type and size, fabrication method, and characterization method. Data items about material design and incorporation of the BG included dressing category, type of dressing/matrix, purpose of the dressing, addition of other bioactive components, method for BG incorporation, physical properties/characterization techniques, and ion release. Data items of the in vitro results included type of cells, cytocompatibility/biocompatibility, cell attachment, antimicrobial properties, angiogenesis, and other findings. Data items of the in vivo results included animal model, cohort size and characteristics, different groups, characteristics of inflicted wound, wound follow-up, wound healing, histological analysis and immunofluorescence analysis, angiogenesis, and other findings. In case no numerical values on in vivo wound healing results were available, a tool (http://www.graphreader.com/) (accessed on 3 August 2023) was used to approximate numerical values from the figures. 

If additional information was required or essential information was missing, the first author was contacted. In case of no response, a follow-up email was sent one week later. However, if there was no answer within 4 weeks, a third attempt was not made. 

### 2.5. Outcomes and Synthesis Methods 

Risk of bias assessment was not conducted, as the studies included in this review were limited to in vitro and in vivo animal studies, for which risk of bias assessment is not required. GRADE (Grading of Recommendations, Assessment, Development, and Evaluations) was considered for evidence assessment. However, the articles included in this review do not permit one to perform a GRADE analysis, since the study did not focus on clinical data but rather on fundamental research. 

The primary aim of this systematic review was to create an overview of the different bioactive-glass-based biomaterials and their effect in diabetic wound healing in vitro and in vivo. Therefore, a formal narrative synthesis was conducted to identify the different production methods, morphologies, and compositions of bioactive glass formulation and their properties. The in vitro results focus on cell viability, ion release, and antimicrobial properties, and in vivo results on diabetic wound healing in an in vivo animal model. Studies were grouped based on material technology as hydrogel, electrospun fibers, and miscellaneous, including scaffolds, ointments, bioactive glass extracts, and composites. These results are displayed in tables and no statistical analysis or meta-analysis was performed. 

## 3. Results

### 3.1. Study Selection

The studies for this systematic review were identified using four different databases: PubMed, Scopus, Web of Science, and Embase Ovid, as seen in the PRISMA flow diagram (Figure 2). This flow diagram shows all of the process, from identification to full text screening, and it is explained next. From the databases used, a combined 105 articles were obtained, 66 of which were removed since they were duplicates, leaving 39 records to screen. Two screenings were performed: in the first screening, the articles were screened by title, abstract, and keywords, and the second one consisted of full text screening. After the first screening, 14 articles were excluded. The second screening further excluded six articles. The reasons for why articles were removed after the full text screening were: wrong publication type (n = 3), wrong animal model (n = 2), and wrong tissue (n = 1). More details on the articles excluded after full text screening can be seen in Table 1.

### 3.2. Study Characteristics

Nineteen articles were included in this systematic review, 89% (seventeen studies) of which were from China; the remaining studies were from India and Egypt. All articles included an in vivo diabetic animal model, 63% (12 studies) consisted of diabetic rat models, 32% (6 studies) used diabetic mouse models, and 5% (1 study) of the articles used a diabetic rabbit model.

Sixteen of the included studies performed in vitro work. Nine of the studies used HUVECs to assess the cytocompatibility of the material, four of them used L929 fibroblasts, and two used human dermal fibroblasts (HDFs). The remaining studies used one of the following cells: endothelial cells, endothelial progenitor cells (EPCs), human acute monocytes leukemia THP-1 cells (THP-1 cells), mouse aortic endothelial cells (MAECs), human neonatal foreskin epidermal keratinocytes, and mouse macrophages (RAW 264.7 cells, ATCC, USA).

Nine of the studies evaluated angiogenic properties in vitro using a tube formation assay or VEGF expression, and sixteen of them evaluated angiogenic properties in an in vivo animal model. Five of the studies analyzed the antimicrobial properties of their material in vitro, and none of them performed in vivo antimicrobial studies. All of the studies received financing from different institutions; however, none of them declared to have conflicts of interest. A more detailed view of this information can be found in Table 2. 

### 3.3. Overview of the Bioactive-Glass-Based Materials 

The materials developed in the articles included in this systematic review include hydrogels, fibers, ointments, composites, and scaffolds. This review will study these materials in three different categories: hydrogels, fibers, and miscellaneous (Vaseline-based ointments, extracts, multilayer composite).

### 3.4. Hydrogel-Based Materials Containing Bioactive Glass

Seven of the nineteen studies reviewed consisted of hydrogel-based dressings incorporating bioactive glass. Table 3 describes these materials in detail. The main purpose of the materials is to aid diabetic wound healing, but each one of them has a different approach for achieving this; this approach is addressed in the purpose of the dressing column. More information on the synthesis of the material can be found in Appendix B. 

#### 3.4.1. In Vitro Studies

##### **Cytocompatibility** 

The cytocompatibility of the different hydrogel-based materials was studied for all of the materials presented in Table 3. Chen et al. (2021) demonstrated using a CCK-8 assay that the Ce-doped BG/gelatin methacrylate (n Ce-BG/GelMa) composites supported the growth and proliferation of HUVECs and L929 (mouse fibroblasts) [41]. Li et al. (2022) tested the biocompatibility of SA/BG-MMP9-siNP and showed that HDFs maintained good biocompatibility after a 3-day timeframe [49]. Kong et al. (2018) observed that the BG extracts had no cytotoxic effect on HUVECs at day 0, 1, 3, and 5; DFO also did not affect cell viability. The cell migration assay indicated that the BG+DFO combination had the strongest effect in promoting HUVEC migration in comparison to BG extracts or DFO alone [46]. Hu et al. (2021) evaluated the effect of a diabetic environment in cell proliferation in vitro; for this purpose, they treated HUVECs with different glucose concentrations. CCK-8 analysis showed that the high glucose (HG) inhibited HUVECs viability at a dose above 25 mM. The EdU cell proliferation assay showed that HG suppressed HUVECs in a dose-dependent manner. However, this study focused on the effect that exosomes had as part of the dressing containing bioactive glass, so the effect of the bioactive glass itself was not isolated [43]. Hu et al. also showed using live/dead staining that the SIS/MBG@Exos hydrogel scaffolds group was significantly higher than that in the SIS@Exos group, meaning that mesoporous bioactive glass (MBG) could positively affect cell viability [43].

Li et al. (2020) studied the effect of PA (PAB-0), PAB (PAB-3), and PABC (PABC-3) scaffolds on EPCs proliferation using EdU and CCK-8 assays [48]. After 48 h of co-culture with the PABC scaffold, the cell viability and proliferation were significantly enhanced compared with the PAB scaffold and control, which indicates that Cu might have an effect in cell proliferation. EPCs proliferation of the PAB scaffold group was also significantly higher than the control; the PABC group still had advantages in cell viability and proliferation, with a higher OD value of live cells and more EDU-stained positive cells.

Ma et al. (2020) found no effect in the proliferation of RAW264.7 after being in contact with SA/BG hydrogels for the first 3 days. After a 7-day culture, the SA/BG hydrogel with 2% and 3% BG showed inhibitory effects on cell proliferation. The hydrogel containing 1% BG did not inhibit cell proliferation. These results suggest that a high BG content could inhibit cell viability [52].

Zhu et al. (2022) studied the effect of bioactive glass extracts containing different ratios of magnesium. They evaluated the following groups: 0MgBG, 2MgBG, 5MgBG, and 10MgBG [56]. The results showed no obvious difference in the cell density between the extract groups and the blank group after incubation for 1, 3, and 5 days, demonstrating that the BG extracts were not toxic for cells. After co-culturing with the hydrogels for 24 h, there were few dead cells and the majority of L929 cells remained alive, suggesting that the hydrogels possessed good cell compatibility [56]. 

These studies collectively emphasize the cytocompatibility and potential benefits of bioactive glass incorporated into a hydrogel dressing for various cell types, highlighting their suitability for wound healing applications, specifically from stages II–III (Figure 1).

##### **Angiogenesis** 

Six out of the seven hydrogel-based material studies analyzed the angiogenic properties in vitro. Four of them used the tube formation assay to achieve this. Two of them evaluated the response of growth factor VEGF and angiogenic receptors KDR and eNOS as well as CD31 expression [52,56]. The evaluation of the angiogenesis in vitro and more details about the assay can be found in Table 4.

From the information presented in Table 4, it can be concluded that all the materials possessed angiogenic properties in vitro. However, there is no quantification of the angiogenic properties in comparison to the control groups. 

##### **Antimicrobial Properties** 

The antimicrobial properties of several hydrogel-based materials are summarized in Table 5. It is relevant to highlight that only three out of the seven articles studied the antimicrobial properties of their material, this being a missed opportunity considering that some BG compositions are known to have antimicrobial properties [57] and DFUs are usually infected [58]. Chen et al. (2021) demonstrated that nCe-BG/GelMA (n = 0, 2, 5) reduced colony numbers of both Gram-positive and Gram-negative bacteria. The survival percentage of bacteria decreased as the BG concentration increased. The composite hydrogel using 5 mol% Ce, called 5/G, had a survival rate of (9.59 ± 0.43)% for *E. coli* and (32.81 ± 4.18)% for *S. aureus* [41]. 

PEGDA-based hydrogel containing bioactive glass nanoparticles with and without copper produced by Li et al. (2020) had minimum inhibitory concentrations (MICs) of 270 µg/mL (PABC) for *S. aureus* and 125 µg/mL for *E. coli*, while no obvious antimicrobial activity was observed for PAB. The PABC-3 (3% Cu) scaffold had excellent antimicrobial activity against *S. aureus* and *E. coli* [48]. To further verify the robust antimicrobial effect of the PABC-3 scaffold, the recyclable antimicrobial ability of the PABC scaffold was investigated. Similar concentrations and volumes of bacteria to previous experiments were selected and added at 0, 3, and 6 h. For three consecutive additions of bacteria, significantly high bacterial growth was found in the PAB-0 and PAB-3 group; however, the survival rate of the bacteria remained less than 1% in the PABC-3 group, suggesting their excellent recyclable antimicrobial effect. The authors believe that the released Cu^2+^ in the PABC-3 scaffold was probably the main reason for the long-lasting antimicrobial effect. The amount of Cu^2+^ released after 6 h was 1.07 mg/L, which is below the upper daily Cu^2+^ intake level (10 mg) for adults of both genders between 19 and 70 years old [59]. The cytocompatibility assays performed using endothelial progenitor cells demonstrated that the amount of Cu^2+^ used was not toxic for the cells [48].

The spread plate method was used to investigate the antimicrobial properties of HQ, HQB0, and HQB10. The tested hydrogels rarely had bacterial colonies, suggesting that they possessed excellent antimicrobial properties. The killing rate for *S. aureus* is 95.87% (HQ), 97.99% (HQB0), and 95.08% (HQB10), and 92.37% (HQ), 97.17% (HQB0), and 98.88% (HQB10) for E coli. They attribute the antimicrobial properties to the QCS as it has a strong positive charge of amino and quaternary ammonium groups, which can destroy bacterial cell membranes and disturb bacterial metabolism by electronic interaction [56].

##### **Macrophage Differentiation** 

The effect of macrophage differentiation was only studied by two of the articles in the hydrogel group. Li et al. (2022) studied the effect of macrophage polarization; they noticed that BG stimulated differentiation from phenotype M1 to M2, meaning that the macrophages go from the inflammatory phenotype to the proliferation phenotype, promoting the transition from stage II to stage III in the wound healing process (Figure 1) [49].

Ma et al. (2020) noticed that SA/BG hydrogels stimulated RAW 264.7 cells to differentiate from the M1 phenotype to the M2 phenotype. Among all SA/BG hydrogels, SA-2% BG hydrogel had the highest stimulatory effects on the M2 polarization of RAW 264.7 cells [52]. Interestingly, SA-3% BG could stimulate 63% of RAW 264.7 cells to polarize into M1 phenotypes, which was significantly higher than that in other groups. However, the SA/BG hydrogels with 1% and 2% BG had no effect on reducing the M1 phenotype cells compared with DMEM. Therefore, 2% BG (*w*/*v*) was the suitable amount in the hydrogel to stimulate M2 polarization of RAW264.7 [52].

#### 3.4.2. In Vivo Animal Studies

All hydrogel-based material studies examined the wound healing and angiogenic properties in vivo (Table 6). The 5/G dressing produced by Chen et al. (2021) demonstrated a significantly better and accelerated wound healing in diabetic rats compared to the control group treated with only gauze [41]. The 5/G group showed an accelerated formation of granulation tissue, re-epithelization, collagen deposition, and angiogenesis, resulting in a better skin reconstruction. In general, the hydrogel groups showed less inflammation compared to the control. The biosafety of the material was assessed; due to the presence of Ce^4+^, the analysis of the liver and kidney function remained in normal ranges, indicating that Ce^4+^ was not harmful to the rats. 

The investigation conducted by Hu et al. (2021) showed no significant differences between the groups, although the SIS/MBG@Exos hydrogel demonstrated a smaller wound size and relatively faster healing compared to the untreated control group, though not statistically significant [43]. Nonetheless, the SIS/MBG@Exos group exhibited a significantly smaller wound length and improved wound perfusion as measured by Doppler compared to the control group. Histological analysis revealed that the SIS/MBG@Exos hydrogel significantly promoted re-epithelialization, as evidenced by H&E staining, and showed a significant increase in collagen III deposition compared to the control group as shown by Masson staining. Additionally, the SIS/MBG@Exos group exhibited a significant increase in the VEGF and CD31 positive area, indicating enhanced angiogenesis [43]. 

SA-BG/DFO produced by Kong et al. (2019) had significantly better wound healing compared to the untreated control group and SA-BG at day 20 [46]. The new dressings had better collagen deposition, in which SA-BG/DFO was the best ordered and had a higher resemblance to the normal skin structure. Re-epithelization was highest in the SA-BG/DFO group compared to other groups (92.6% vs. 60.3%, 80.3%, and 77%). Lastly, SA-BG/DFO had the most mature blood vessels and angiogenesis at day 12, followed by a significant reduction at day 20, indicating quick and extensive angiogenesis in the early stage of wound healing [46]. 

The PABC scaffold made by Li et al. (2020) also showed significantly faster wound healing, with almost complete re-epithelization and the highest collagen deposition at day 20 in comparison to the untreated control group and other dressings [48]. The diabetic wounds of both the PABC- and PAB-treated groups consisted of new granulation tissue and had a more organized fiber density compared to PA and the control. PABC had the best angiogenic properties, with increased neovascularization in the early stage of wound healing [48]. 

All the hydrogel dressings produced by Li et al. (2022) had better wound healing than the untreated control group [49]. The rsiNP-BA/SA hydrogel had a complete re-epithelization of the wounds, showed the most ordered collagen fiber deposition, and demonstrated the strongest downregulation of matrix metallopeptidase 9 (MMP-9). In addition, all dressings had better neovascularization than the control, but the BG/SA group had the most vessels as shown in the SMA-alpha and CD31 staining, indicating that siNP does not have an additional value to neovascularization in diabetic wound healing [49]. 

Ma et al. (2020) demonstrated that wound healing in the SA/BG-SA_CM_-PLGA and SA/BG-SA_CM_-PLGA_PFD_ groups (when using 2% BG for SA/BG) was almost 100% and significantly better than the untreated control group and the other two new dressings [52]. Additionally, the abovementioned dressing formed neo-epidermis in the early stages of wound healing, as well as increased granulation, collagen I and III deposition, and re-epithelization. All hydrogel groups showed increased angiogenesis at day 12, of which the SA/BG-SA_CM_-PLGA_PFD_ had the highest angiogenic properties and highest M2:M1 macrophage ratio. In the later stages of wound healing, the BG-SA_CM_-PLGA and SA/BG-SA_CM_-PLGA_PFD_ groups had a significantly reduced number of mature blood vessels and more collagen I in comparison to collagen III, indicating more mature collagen compared to the other treatments [52]. 

Zhu et al. (2022) demonstrated that HQB10 had a significantly better and faster wound healing capacity compared to the other dressings and the untreated control group [56]. All hydrogel groups showed reduced inflammation and higher ECM and collagen deposition and compact granulation tissue of the wound. HQB10 had the most fibroblasts, the densest ECM, and the most organized collagen deposition and re-epithelization of the wound. Furthermore, HQB10 had the best angiogenic properties, with a quick and extensive angiogenesis in the first 7 days [56]. 

Overall, it can be concluded that the addition of BG to the hydrogel results in better wound closure when compared to the controls, ranging from approximately 88% to 100%, as well as quick and extensive functional angiogenesis at the early stages of wound healing, as shown with Doppler analysis. In addition, these findings indicate that the addition of BG also improves wound remodeling. 

### 3.5. Electrospun Fibers and Scaffolds Containing Bioactive Glass

The second largest category of dressings found in this study was the ones synthesized using electrospinning. Electrospinning was used to form different fiber sizes and scaffolds; a description of the materials and their purpose can be found in Table 7, and more details on how the materials were synthesized can be found in Appendix B. 

The main purpose of all of the synthesized dressings is to help diabetic wound healing, and each one of the materials has a different approach to achieve this. These materials mainly rely on the similarity of their structure to the ECM and ion release (Si^4+^) to accelerate the wound healing process. 

#### 3.5.1. In Vitro Studies

##### **Cytocompatibility** 

Li et al. (2017) noticed that BG nanocoatings on the scaffold significantly increased the cell viability of HUVECs at days 3 and 7 [47]. Many stained cells on BG/PEM exhibited well-defined distinct microfilaments and cytoskeletons in comparison to those on PEM, where the cells were much fewer and not well spread. SEM images showed that both BG/PEM and PEM supported the attachment of HUVECs and that the cells were not only closely attached to the surface of the nanocomposite but also in the inner part of the microfibers. Chen et al. (2019) studied a chitosan/PVA/PVAnBG scaffold and saw that cell proliferation increased as the concentration of nanobioactive glass particles increased [40]. The 40% nBG (nBG-TFM) gave the best fibroblast (L929) proliferation results. SEM and fluorescence microscopy showed that cells were adhered to the membrane. 

Gao et al. (2017) used endothelial cells to assess the cytocompatibility of their CPB and CP scaffolds [42]. After 3 days of culture, endothelial cells exhibited a significantly higher number of filopodia and larger spreading on CPB scaffolds compared to CP alone, and high cell proliferation was also seen on CPB scaffolds after 5 days. These results suggest that CPB is a good candidate for cell spreading and proliferation.

Jana et al. (2022) evaluated the cytocompatibility of their electrospun copper- and cobalt-doped bioactive glass–fish dermal collagen mat using human dermal fibroblasts [44]. The mats proved to be cytocompatible and hemocompatible with rabbit RBC. Flattened fibroblasts on each of the microporous mats can be seen by SEM, indicating the efficient cell–ECM interactions. It can be said that the web-like features of the microfibers provide pockets for cellular entrapments while preserving cellular morphology, which affirms the cytocompatibility features of these electrospun mats.

Jiang et al. (2020) used HUVECs to test the biocompatibility of their mussel-inspired composite scaffold [45]. The cells cultured on the PDA/PM and BG/PDA/PM scaffolds showed a well-adhering and spreading morphology with cytoskeletal extensions, and the number of adherent cells was significantly higher than that of the PM group. The migration of HUVECs cultured with different scaffolds showed that the number of migrated cells in the BG/PDA/PM group was much higher than those of the other groups, which exhibited the strongest promoting effects of BG/PDA/PM scaffolds on cell migration. The proliferation rate of HUVECs cultured on both PDA-coated scaffolds was significantly higher than that of the pure scaffold at days 3 and 7. These results show that the bioactive glass has a positive effect on HUVECs migration and proliferation.

Zhang et al. (2020) used HUVECs to study cell proliferation and migration [53]. The proliferative status of the cells on days 1, 3, and 7 demonstrated that the PG scaffold, especially the BG@PG scaffold, could promote the proliferation of endothelial cells, which indicates that the material is biocompatible. From the scratch test, it can be seen that, after 24 h of culture, the migration rate of the cells in the BG@PG group reached 72.2%, which was significantly higher than that of PG (61.5%), UPG (54.9%), and UP (56.4%).The migration rate of cells was also significantly improved in the PG and BG@PG scaffolds, which may be due to the micropattern structure that could provide a sufficient adhesion site for cell growth, while the Si ions released from the scaffold could increase the proliferation rate of cells. Thus, the scratch width in the PG group was significantly reduced compared to the UPG group after 24 h of incubation, while the scratch of the BG@PG group almost disappeared.

Researchers have explored different material solutions to aid diabetic wound healing using electrospinning techniques. Three of the studies used HUVECs to analyze cytocompatibility, one of them used endothelial cells, and one used human dermal fibroblasts. All the materials in this section were biocompatible and promoted cell attachment and cell proliferation, showing that electrospun fibers/scaffolds are a promising dressing solution to aid wound healing. 

##### **Angiogenesis** 

Li et al. (2017) evaluated the angiogenesis differentiation of HUVECs on BG/PEM and PEM by RT-PCR analysis. The gene expression level of VEGF and eNOS of BG/PEM were significantly higher than those on PEM. BG/PEM stimulated angiogenesis in HUVECs by improving the expression of angiogenesis-related genes, including VEGF and eNOS [47].

Chen et al. (2019) studied the effect of nBG-TFM in the expression of bFGF and VEGF by rt-PCR in HDF. They noticed that nBG-TFM significantly promoted a higher expression of bFGF and VEGF (in comparison to the control), which are related to angiogenesis and essential for tissue regeneration [40].

Gao et al. (2017) noticed a large amount of positive staining of CD31 in the CPB and CP groups and that none of the activated CD31 was found in the control group. These results suggest that the CPB scaffolds can significantly enhance the angiogenesis marker expression of CD31 [42].

Jiang et al. (2020) used a tube formation assay and q-rt PCR to evaluate angiogenesis in vitro. After 6h, the number of nodes in the BG/PDA/PM group was significantly higher than in other groups. The PDA and BG/PDA/PM groups showed a higher increase in the expression of angiogenic genes than PM [45]. Both the tube formation assay and the VEGF expression suggest that bioactive glass can effectively promote angiogenesis in vitro. However, it is important to mention that none of these assays can show the formation of an actual functional network.

##### **Antimicrobial Properties** 

Chen et al. (2019) observed that, without the chitosan component, the PVA-nBG membrane exhibited almost no antimicrobial activity toward Gram+ and Gram− bacteria [40]. On the other hand, all kinds of membranes containing chitosan exhibited inhibitory effects on bacterial growth. Among them, trilayer nanofibrous membranes with the highest chitosan content in the sublayer showed the best inhibitory efficiency. It could also inhibit *Pseudomonas aeruginosa* growth, which are the most common bacteria isolated from chronic wounds.

Sharaf et al. (2022) studied the antimicrobial properties of their materials using an agar well diffusion method. BGNP containing a CA mat had superior antimicrobial properties as compared to un-fabricated CA mats, with a wide range of activity against the Gram-positive (*Bacillus cereus*, *Bacillus subtilis*, and *Staphylococcus aureus*) and the Gram-negative (*Salmonella typhimurium* and *Escherichia coli*) bacteria [54]. It is worth noting that the BGNP-containing CA nanofibers were also effective against *Staphylococcus aureus* and *Escherichia coli* strains.

#### 3.5.2. In Vivo Animal Studies

All studies that examined wound healing and angiogenic capacities of electrospun fibers and scaffolds in vivo are described in Table 8.

Chen et al. (2019) showed that the addition of BG in nBG-TMF resulted in a faster healing process, with significant differences in wound healing between the nBG-TMF and control group, treated with petrolatum gauze, at day 7 and 14 [40]. Nevertheless, TFM was described in the results section, but no values or graphs were provided by the authors. Histological analysis showed enhanced skin regeneration with regard to collagen alignment and the formation of skin appendages, which was comparable to healthy skin. Furthermore, VEGF and TGF-beta were significantly upregulated in nBG-TMF, indicating that the addition of BG improves angiogenesis in diabetic mice. Lastly, inflammatory cytokines were significantly more downregulated in nBG-TMF compared to the control and TFM. In general, Chen et al. (2019) demonstrated that the addition of BG to the electrospun dressing achieved accelerated and enhanced wound healing in diabetic mice [40]. 

The BG/PEM composite of Li et al. (2017) showed significantly better and faster wound healing at day 7 and 13 compared to the PEM and control groups [47]. The addition of BG contributed to a more extensive and continuous formation of the epidermis, with thick collagen fibers and extensive collagen deposition. Additionally, the wounds treated with BG/PEM showed a significantly increased CD31 and VEGF expression and a higher density of new blood vessels at day 15 compared to the control and blank. Lastly, collagens I and III were upregulated in BG/PEM in comparison to the control. The abovementioned suggests that BG/PEM could enhance diabetic wound healing and improve the quality of the newly formed skin. 

The CPB dressing of Gao et al. (2017) significantly increased the wound healing rate compared to the CP dressing or the untreated control group [42]. At day 7, diabetic wounds treated with CPB showed granulation tissue, an increased number of mature blood vessels, fewer inflammatory cells, and fibroblasts. At day 14, re-epithelization and more organized collagen depositions were present in the diabetic wounds treated with CPB. In addition, the area of scar tissue was reduced in the CPB group compared to the CP and control. 

Jana et al. (2022) was the only study that used rabbits, and the bioactive mats were replaced every four days in comparison to the other studies included in this review [44]. All bioactive mats showed an increased healing rate, downregulated inflammation, and granulation tissue compared to duoderm and the untreated control group. There were no significant differences in wound area between duoderm and the bioactive mats after the treatment period. Furthermore, Fcol/CuCoBG showed the highest hydroxyproline content, which is important for collagen deposition and ECM remodeling, as well as the most granulation tissue and the most complete re-epithelization compared to the other bioactive mats or control dressings. All bioactive mats showed an increased CD31 expression and number of mature blood vessels at day 7 and 14 compared to the control dressings. With regard to the bioactive mats, Fcol/CuBG, Fcol/CoBG, and FcolCuCoBG were superior to the Fcol/BG treatment, which could validate the addition of Cu- and Co-ion-doped BG. 

The BG/PDA/PM group of Jiang et al. (2020) had significantly better wound healing compared to the control (patterned scaffold) and other new dressings [45]. PDA/PM and BG/PDA/PM had a downregulation of inflammatory cytokines. Wounds treated with BG/PDA/PM presented a significantly increased CD31 expression in the first 7 days and downregulated CD31 expression at day 15, indicating quick and extensive angiogenesis in the early stages of wound healing. The BG/PDA/PM group had a remarkably higher and more mature blood vessel density compared to the other groups. Both BG/PDA/PM and PDA/PM groups showed re-epithelization, of which BG/PDA/PM had a more mature epithelial structure and the thickest neo-epidermis. Lastly, BG/PDA/PM had increased collagen deposition, which was more organized, and, at day 15, expression of Col I was higher and Co III expression was decreased in the BG/PDA/PM group compared to the other dressings. 

The PG and BG@PG group of Zhang et al. (2020) had a significantly smaller wound area at day 7 than UPG and the untreated control group [53]. However, on day 14, the BG@PG group showed significantly superior wound healing in comparison to all groups. The H&E staining showed a complete and continuous neo-epidermis in the PG and BG@PG group. CD31 and VEGF expression was significantly higher in the BG@PG group at day 7 and was decreased at day 14, indicating the quick and extensive angiogenesis in the early wound healing stage. In addition, the collagen deposition, organization, and density were significantly higher in the BG@PG group in comparison to the other dressings. 

Sharaf et al. (2022) suggest that the 3%-BGNP-incorporated CA nanofiber significantly enhanced the wound healing rate in diabetic rats in comparison to an unfabricated CA nanofiber [54]. No results were available on the fibroblasts, granulation tissue, inflammatory cytokines, and collagen of angiogenic factors relevant to wound healing. No complications or side effects were seen in the in vivo experiment, indicating its biocompatibility. 

To summarize, these findings suggest that electrospun fibers with BG enhance diabetic wound healing in vivo in comparison to the controls, ranging from 78% to 100% wound closure, with extensive angiogenesis in the early stage of wound healing, as well as improved wound remodeling. 

### 3.6. Miscellaneous: Ointments, Bioactive Glass Extracts, and Composites

Besides the hydrogels and fiber-based materials, this review covers other varieties of materials classified under the term miscellaneous since they do not fit into one specific category. In Table 9**,** a detailed description of these materials can be found; some of them are ointments or extracts, so do not correspond to a dressing, and are more of a bioactive-glass-based treatment. More details on the fabrication of these materials can be found in Appendix B. These materials aid diabetic wound healing in diverse ways as seen in Table 9.

#### 3.6.1. In Vitro Studies

Tang et al. (2021) used human neonatal foreskin epidermal keratinocytes to study the biocompatibility of their bioactive glass ionic extracts [55]. After cells had adhered, the culture medium was replaced by the diluted BG extracts, and the keratinocyte culture medium without the BG extracts was used as the control. Cells were harvested at days 1, 3, and 7 for the CCK-8 assay; the medium was refreshed every 3 days. In the first 3 days, the cells cultured in the BG extracts exhibited a similar growth pattern as in the control. This trend continued for the next four days for 1/80 and 1/320. However, the OD value of the 1/20 group was much lower than the control, indicating that BG extracts diluted at the ratio of 1/20 suppressed the proliferation of keratinocytes. The cell migration of keratinocytes was studied through the scratch assay, and a large number of cells were observed in the scratch area of each group after 24 h. Quantitative statistical analysis showed that the migration rates of the 1/20 group (48.30 ± 2.79)%, the 1/80 group (58.01 ± 1.49)%, and the 1/320 group (53.6 ± 5.80)% were close to that of the control group (55.75 ± 5.28)%. There was no statistical difference between the groups. However, it was noted that cells in the scratch area of the control group were scattered, with no visible cell–cell contacts between each other, suggesting a single-cell migration. On the contrary, cells cultured in the BG extracts migrated collectively like sheets. Most cells retained their cell–cell contacts when they moved forward.

Xie et al. (2019) used mouse macrophage cells (RAW.264.7) to test biocompatibility [25]. The concentrations of BG particles used in the in vitro study were 5, 20, 50, and 100 mg/mL. Macrophages cultured without BG particles were used as controls. At low concentrations (20 mg mL^−1^), BG promoted the proliferation of macrophages with increasing concentration without compromising the viability; in contrast, at high concentrations (450 mg mL^−1^), BG inhibited cell proliferation, and the inhibition intensity increased with increasing concentration. The bioactive glass ointment also showed an effect on macrophage differentiation. The macrophages with 20 mg/mL BG particles showed mild inflammatory responses (M2/M1 = 0.8) at first and then switched to the M2 phenotype (M2/M1 = 1.4), while more M1 macrophages were persistently detected with the 100 mg/mL treatment. 

Both previous studies show that lower bioactive glass concentrations promote cell viability and proliferation while higher concentrations can compromise cell viability. The cells used for the studies corresponded to keratinocytes and macrophages. Xie et al. (2019) also demonstrated that a BG concentration of 20 mg/mL could promote macrophage polarization from an M1 phenotype mainly associated with the pro-inflammatory response to an M2 phenotype associated with anti-inflammatory responses [25].

#### 3.6.2. In Vivo Animal Studies

In the last category, miscellaneous, the wound healing capacities of five different BG-based materials are evaluated, those being one composite, three ointments, and one extract. An overview of the miscellaneous treatments is depicted in Table 10. 

Bao et al. (2020) investigated the diabetic wound healing properties of composites in vivo, of which the 20BG ointment had significantly faster and better wound healing at all time points compared to the control or 0BG dressing, indicating the advantage of BG in diabetic wound healing [39]. In this study, Bao et al. also investigated the water absorption capacity of the materials, which was not affected by the addition of bioactive glass [39]. Both composite dressings (20BG, 0BG) significantly inhibited tissue edema in comparison to the control. Additionally, diabetic wounds treated with 20BG had a significantly higher amount of blood vessels than 0BG and the control. The latter was consistent with the CD31 expression. 

Xie et al (2019), Mao et al. (2014) and Lin et al. (2012) studied the diabetic wound healing properties of ointments in vivo [25,58,59]. In the study of Mao et al., ointment II showed significantly better wound healing on day 14 compared to the control and ointment I, IV, and V [51]. Nevertheless, the paper shows an inconsistency in referring to the results of ointments II and III. Therefore, we interpreted the results on the basis of Table 1 in the paper of Mao et al. [51]. On day 21, all wounds were covered with a new epidermis, but wounds treated with saline and Vaseline were still open. Diabetic wounds treated with ointment II had large new granulation tissue, fewer inflammatory cells, and more fibroblasts. With regard to angiogenic properties, the highest number of blood vessels and highest VEGF expression were also seen in wounds treated with ointment II. These results indicate that, for this particular case, a low concentration of BG (5%) is better than high concentrations of BG for diabetic wound healing. 

The SGBG-58S ointment of Lin et al. (2012) had the fastest wound healing compared to all groups [50]. The wound healing rate at 14 days was significantly better in the SGBG-58S and NBG-58S group. The mice treated with BG had an increased healing rate of 50% wound closure within the first six days and approximately 90% within ten days, with thick granulation tissue comparable to normal skin. Furthermore, more inflammatory cells, fibroblasts, and blood vessels were present in the H&E staining of mice treated with BG than the control. At day 14, the skin of diabetic mice treated with BG was almost healed, collagen was present, and new epithelium was formed. Lastly, VEGF secretion was highest in the BG-treated mice, of which SGBG-58S and NBG-58S dressings showed the highest number of newly formed blood vessels at day 7, and VEGF expression decreased at day 14, indicating quick and extensive angiogenic properties in the early wound healing stage. 

Xie et al. (2019) showed that an LBG could significantly improve diabetic wound healing compared to HBG and the control [25]. In the first 7 days, the wounds of the LBG group were the smallest and, by day 14, the wounds of the LBG group were almost completely closed, whereas the wounds of the HBG and control group were still open. Interestingly, the wound healing was decelerated in the HBG group and accelerated in the LBG group, indicating that BG concentration affects wound healing. A high concentration of BG results in a later and persistent inflammatory state of the M1 macrophages, which could explain the suboptimal wound healing. On the other hand, the LBG group had more M1 macrophages in the early stage (day 7), but no M1 macrophages and increased M2 macrophages at day 14, suggesting that LBG can initiate M1-to-M2 transition, allowing the re-epithelization process to occur. In addition, keratinocyte migration was only observed in the LBG group, which is essential for the re-epithelization process. 

Lastly, Tang et al. (2021) used BG extracts for diabetic wound healing assessment in vivo [55]. The rats treated with 1/20BG had a significantly more reduced wound area at day 7 and day 14 compared to the control and blank. Moreover, diabetic wounds treated with 1/20BG extracts showed stronger K10 and involucrin expression, indicating increased keratinocyte differentiation and upregulation of ZO-1 and claudin-1 in keratinocytes. The latter indicates that the addition of BG could promote the differentiation of keratinocytes and enhance tight junction formation in the new epidermis, restoring skin barrier function. 

## 4. Discussion

Wound healing is a complex process consisting of four overlapping stages: hemostasis, inflammatory, proliferative, and remodeling phases, as shown in Figure 1. Each one of these stages is relevant to ensuring that a wound heals properly. The studies included in this systematic review focus their in vitro work mainly on the inflammatory and proliferative phase, while the majority of the in vivo work focuses on the proliferative and remodeling stages. 

Most of the authors tested the cytocompatibility of their material in vitro, all of them obtaining a more favorable result with the BG-based material in comparison to their respective control. The studies used different cell lines, methods, and controls to evaluate cytocompatibility and proliferation, so a quantitative comparison between them cannot be made. We consider that the cells used to study cytotoxicity should be selected carefully and accordingly to the stage of wound healing that the material aims to support. If the BG-based dressing is targeting the proliferative stage, then the most adequate cells to use would be fibroblasts and keratinocytes, as well as macrophages, as these are the ones involved in this phase [61]. In the in vivo studies, no side effects or complications with the materials were specifically mentioned, which could be interpreted with caution as them not being observed. Only Chen et al. (2021) reported no cytotoxic events in their hydrogel dressing [41]. 

The effect of diabetes was not considered in vitro by most of the authors. Hu et al. (2021) tried to consider the effect of diabetes by increasing the glucose concentration in the medium; they noticed that high glucose suppressed HUVECs in a dose-dependent manner [43]. This result illustrates the importance of evaluating the effect of high glucose when assessing material–cell interactions in vitro. Nevertheless, all in vivo studies included a diabetic animal model to assess the wound healing process. Overall, all studies showed that the addition of BG in their respective materials showed a faster wound healing process in vivo compared to their respective controls or other dressings without BG. In some studies, ion-doped BG was added to the dressing—ions such as Cu^2+^ to the PABC dressing of Li et al. (2020)—which significantly improved wound healing in comparison to the PAB dressing in which only BG was added [48]. 

### 4.1. Inflammatory Stage 

The inflammatory phase was not the direct target of any of the materials included in this review; however, some of them did study the polarization of macrophages from the pro-inflammatory M1 phenotype to the reparative M2 phenotype. It was seen in the hydrogels of Li et al. (2022) and Ma et al. (2020) and the ointment of Xie et al. (2019) that bioactive glass incorporated into their respective materials could stimulate macrophage polarization from the M1 phenotype to M2 phenotype in vitro, where this change in phenotype promotes the wound healing process [25,43,44]. Moreover, the in vivo experiment by Xie et al. (2019) and Ma et al. (2020) showed that BG can initiate M1-to-M2 transition, allowing the proliferative and remodeling stage of wound healing to occur [25,44]. The latter improves the wound healing quality, with a better resemblance to normal skin. In general, all authors that studied the inflammatory cytokines in vivo found a significant decrease in inflammatory cytokines. Importantly, Ma et al. (2020) found that the best wound healing was in correspondence with higher M2:M1 macrophage ratios [52]. Xie et al. (2019) showed that BG concentration influenced macrophage polarization in vivo [25]. There is a correlation between the bioactive glass concentration and the ratio of M1 to M2 macrophages. Xie et al. (2019) demonstrated that higher concentrations (over 100 mg/mL) caused an increase in the amount of M1 macrophages, which was confirmed with delayed wound healing in vivo [25]. Therefore, we consider that more studies should be carried out on the effect of bioactive glass in macrophage polarization to determine which concentration range works best to move from the pro-inflammatory (M1) to the reparative (M2) phenotype.

### 4.2. Proliferative Stage

Once the wound enters the proliferative phase (Figure 1), angiogenesis becomes primordial. Angiogenesis refers to the process by which new blood vessels develop from pre-existing vascular structures [62], and it is key for wound healing, as the newly formed capillaries bring oxygen and micronutrients to the growing tissues and remove catabolic waste [63]. Angiogenesis was studied both in vitro and in vivo by 11 out of the 19 papers included, and it was studied only in vivo by 16 out of the 19 papers included. In all of the cases, it was shown that bioactive glass effectively contributed to the formation of new blood vessels. Six of the papers only studied the expression of angiogenic markers in vitro, such as CD31 and VEGF, and did not perform a tube formation assay to evaluate the blood vessel network formation. The angiogenic marker expression was significantly increased in vitro by the materials containing BG, in comparison to the controls, showing the angiogenic-promoting abilities of BG in vitro. With regard to the in vivo results, it can be concluded that the addition of BG significantly increases angiogenesis in comparison to dressings without BG. In fact, VEGF expression and blood vessel density were the highest in the early stages of wound healing (7–14 days) and significantly lower in the later stages of wound healing (14–21 days), indicating a quick and extensive neovascularization that is essential in the early stage. However, the authors did not discuss in which percentage their new material increased blood vessel formation or angiogenic marker expression in comparison to their control. Further studies with quantification methods are warranted to establish conclusive evidence and enable direct comparisons between different dressing compositions. Furthermore, the effectiveness of the newly formed vascular network was not properly evaluated in most in vivo studies. Only Hu et al. (2021) and Li et al. (2020) used a Doppler analysis to evaluate the perfusion of the newly formed blood vessels, which was significantly improved in the dressings with BG [40,42]. Lastly, Mao et al. (2014) found that high concentrations of BG can also inhibit angiogenesis [51]. Subsequently, it is crucial to investigate the optimal BG concentration.

Besides angiogenesis, in vivo results also showed significantly increased fibroblast migration, granulation, and re-epithelization in the early stage of wound healing (day 7–14). Therefore, we can conclude that the addition of BG improves the wound healing time, especially since re-epithelization is the final step of the proliferative stage. 

### 4.3. Remodeling Stage 

The final stage in wound healing consists of the remodeling of the extracellular membrane and scar tissue formation. This stage is not evaluated through the in vitro studies. The quality of the tissue formed was assessed in the in vivo studies by different staining techniques and PCR. Authors found that the collagen deposition, organization, and density were significantly increased in the diabetic wounds treated with dressings that included BG. In addition, PCR results showed a higher collagen-III-to-collagen-I ratio in the early stages, whereas collagen I was significantly increased in the stage after wound healing. Collagen III is characteristic of early phases of wound healing and collagen I is synthesized during the late stages, allowing the tensile strength of the scar tissue to increase gradually due to the alignment of the collagen fibers [13]. Consequently, the diabetic wounds treated with BG-based materials consisted of less scar tissue and had a higher resemblance to normal skin. 

### 4.4. Antimicrobial Properties 

The assessment of antimicrobial properties is also relevant for these materials as 50%–60% of diabetic foot ulcers can develop an infection [64]. Only 5 out of the 19 articles included assessed the antimicrobial properties of their material in vitro, and none of them used a diabetic animal model with infection, hindering the analysis of antimicrobial properties in vivo. Incorporation of bioactive glass in hydrogel-based materials showed increased antimicrobial properties against both Gram-positive and Gram-negative bacteria compared to the control without BG particles. It was found that the reduction in bacterial colony number was directly proportional to the BG percentage present in the material. As for the electrospun scaffolds, Chen et al. (2019) attributed the antimicrobial effect to the presence of chitosan [40], whereas Sharaf et al. (2022) did notice the contribution of BG to the increased antimicrobial properties of their material [54]. 

It is relevant to mention that the antibacterial properties of BG are determined by its composition, size, and concentration [65]. These properties are attributed to an increase in pH and osmotic pressure [57]. Particle size and texture are also relevant since a high specific surface area increases the ability to efficiently release therapeutic ions as well as increase pH and osmolality faster [66]. Some BG formulations may have strong antimicrobial properties and some of them might have none and be focused only on tissue regeneration. In Table 11, the different bioactive glass compositions used in the materials that form part of this systematic review are listed, as well as their antimicrobial properties or lack of them. From the bioactive glasses included in this review, the ones that have proven antibacterial properties correspond to 45S5 and 58S, as well as the formulations by Chen et al. (2021) and Li et al. (2020), and the BG used by Sharaf et al. (2022) [39,42,55].

Bioactive-glass-based materials can be a valuable tool in the fight against antimicrobial resistance (AMR) since, so far, there has not been any evidence shown that bacteria can create resistance to the antibacterial mechanism of bioactive glass [57]. However, further studies are needed to deepen the knowledge of specific antibacterial and antibiofilm activity and improve the use of bioactive glass in the fight against AMR.

### 4.5. Limitations and Further Studies

The review process had several limitations in the data extraction phase. Since material characterization and the in vitro studies vary greatly according to the material prepared, this does not permit a proper comparison of all of the studies. The results for the wound closure percentage were not always mentioned in the text and were just presented in a bar chart; in this case, the data had to be extracted by using image-reading tools to obtain an idea of the values presented. These values cannot be reported with a standard deviation. The same issue occurs with the assessment of in vitro and in vivo results on angiogenic properties. The lack of quantification precluded definitive conclusions. With regard to the in vivo results, the animal model mimics type I diabetes, whereas the most common type of diabetes is type II. In theory, this should not affect the diabetic wound healing results in animal models. However, type II diabetes is often accompanied with metabolic syndrome and other comorbidities in humans, such as obesity and cardiovascular diseases, which should be considered as confounding factors when advancing to clinical trials.

Based on the studies included in this review, we conclude that bioactive-glass-based materials look promising for aiding diabetic wound healing. Further studies are required in which the effect of high glucose is also considered in the in vitro stage and adequate cells are selected for the experiments. We suggest selecting the cell type according to the stage of wound healing that the material intends to accelerate. Studying the effect of BG concentration in macrophage polarization and angiogenesis is also relevant as it allows one to determine a concentration range that is better suited to promoting both the change from M1 to M2 phenotype and angiogenesis. 

When studying wound healing, two different time points should be considered: day 14, in order to look at the different wound healing rates, and day 21, in order to evaluate the wound healing quality. The revision at day 14 is crucial since, by day 21, most of the wounds are closed, so not much can be inferred about the effect of the materials. In addition, the angiogenic properties of the materials should be properly quantified and Doppler analysis is warranted to test the effectiveness of the neovascular network. Future studies should also evaluate the antimicrobial properties of their material in both in vitro and in vivo models. The cost-effectiveness of the newly proposed materials should also be considered to determine if the material is a solution that could be used in clinical practice.

Before advancing BG-based materials to clinical trials, research should focus on in vivo diabetic animal models and study the difference between animals treated with the BG-based material, no treatment, and a control to properly identify the effect of the new material in the wound healing process. Research should also focus on the antimicrobial properties of these materials since, as mentioned previously, they could help in the fight against AMR.

No conclusions can be drawn about the ideal dressing or the additional value of adding ions to the material. Great interest has been given to hydrogel-based materials as they are one of the majority groups in this review; they are also the materials that were studied in more detail, but more research is needed to determine the most ideal dressing for diabetic wound healing. In addition, when finding the ideal dressing for optimal wound healing, it is crucial to take into account factors that are crucial for clinical practice, such as the application of the dressing and the type and location of the diabetic wound. 

## 5. Conclusions

Bioactive glass (BG)-based materials, as discussed in this systematic review, indicate potential benefits by accelerating all of the stages involved in diabetic wound healing and enhancing the overall wound healing quality. Wound healing improves due to the release of therapeutic ions from the bioactive-glass-based materials, increasing angiogenesis, promoting M1-to-M2 macrophage phenotype switching, and, in most cases, avoiding infection due to antibacterial properties. A comparison of the effectiveness of the different materials is not possible since the models used by the studies were not equivalent and the outcome measurements are not standardized. 

It is essential to note that further research on determining the optimal BG concentration to promote angiogenesis, M1-to-M2 macrophage phenotype switching, and wound healing in general, identifying the ideal dressing, and, most importantly, understanding the antimicrobial properties of BG-based materials is warranted in order to eventually advance to clinical trials. In the near future and with more research, a BG-based wound healing material with antimicrobial properties could address the challenges in diabetic soft tissue wound healing and hence greatly assist in the fight against AMR and improve the treatment of diabetic wound infections.

## Figures and Tables

**Figure 2 ijms-25-01152-f002:**
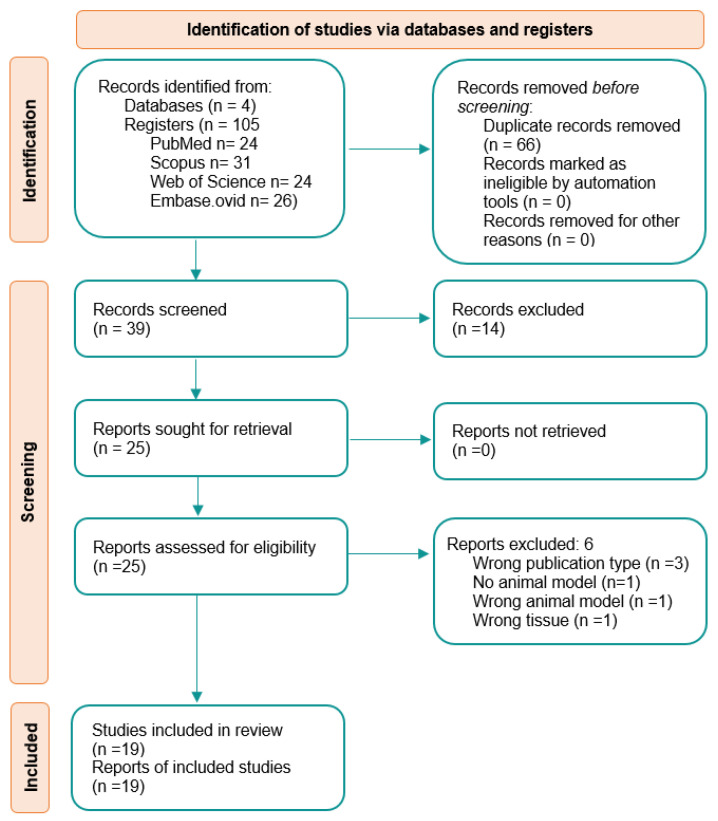
Full text screening illustrated by the PRISMA flow diagram.

**Table 1 ijms-25-01152-t001:** Studies excluded from the systematic review after full text screening.

First Author, Year	Country	Title	Reason for Exclusion
Wray, P., 2011 [33]	United States	Cotton candy’ that heals? Borate glass nanofibers look promising	Wrong publication type: magazine article
Jung, S.B., 2011 [34]	United Sates	Treatment of non-healing diabetic venous stasis ulcers with bioactive glass nanofibers	Wrong publication type: conference abstract
Jung, S.B., 2012 [35]	United States	Treatment of chronic ulcers with bioactive glass nanofibers	Wrong publication type: conference abstract
Deliormanli, A.M., 2014 [36]	Turkey	In vitro assessment of degradation and mineralization of V_2_O_5_ substituted borate bioactive glass scaffolds	No animal model
Wang, Chenggui, 2018 [37]	China	Highly efficient local delivery of endothelial progenitor cells significantly potentiates angiogenesis and full-thickness wound healing	Wrong animal model: no diabetic animal model used
Elshazly, N., 2020 [38]	Egypt	Efficacy of Bioactive Glass Nanofibers Tested for Oral Mucosal Regeneration in Rabbits with Induced Diabetes	Wrong tissue: oral mucosa

**Table 2 ijms-25-01152-t002:** Methodological and study outcome characteristics of articles included in this systematic review.

First Author	Year	Type of Cells	Cytocompatibility	Antimicrobial Properties	Angiogenesis In Vitro		Animal Model	Wound Follow-Up Period (Days)	Angiogenesis In Vivo		
					Tube Formation	Angiogenic Markers			Staining	Gene Expression	Doppler
Bao, Feng [39]	2020	No in vitro work					Diabetic mouse model	13	Y	N	N
Chen, Qingchang [40]	2019	Mouse fibroblast cells (L929) and human dermal fibroblastic cells (HDFs)	Y	Y	N	Y	Diabetic mouse model	21	N	Y	N
Chen, Yue-Hua [41]	2021	L929 and human umbilical vein endothelial cells (HUVECs)	Y	Y	Y	N	Diabetic rat model	21	Y	N	N
Gao, Wendong [42]	2017	Endothelial cells	Y	N	N	Y	Diabetic rat model	21	Y	Y	N
Hu, Yiqiang [43]	2021	HUVECs	Y	N	Y *	N	Diabetic rat model	14	Y *	Y	Y
Jana, Sonali [44]	2022	HDFs	Y	N	N	N	Diabetic rabbit model	21	Y	N	N
Jiang, Yuqi [45]	2020	HUVECs	Y	N	Y	Y	Diabetic mouse model	15	Y	N	N
Kong, Lingzhi [46]	2018	HUVECs early passages 2–7 for experiments	Y	N	Y	Y	Diabetic rat model	20	Y	Y	N
Li, Jinyan [47]	2017	HUVECs	Y	N	N	Y	Diabetic mouse model	21	Y	Y	N
Li, Yannan [48]	2020	Endothelial progenitor cells (EPCs) derived from bone marrow of Sprague–Dawley (SD) rats	Y	Y	Y	N	Diabetic mouse model	21	Y	Y	Y
Li, Ying [49]	2022	Human acute monocytes leukemia THP-1 cells (THP-1 cells), HDFs, and HUVECs. HDFs and HUVECs were used for experiments at passages 7–10 and at passages 3–5, respectively	Y	N	N	N	Diabetic rat model	21	Y	N	N
Lin, Cai [50]	2012	No in vitro work					Diabetic rat model	14	Y	Y	N
Mao, Cong [51]	2014	No in vitro work					Diabetic rat model	21	Y	Y	N
Ma, Zhijie [52]	2020	L929 cells and MAECs	Y	N	N	Y	Diabetic mouse model	18	Y	N	N
Zhang, Pengju [53]	2020	HUVECs	Y	N	N	N	Diabetic rat model	14	Y	Y	N
Sharaf, Samar [54]	2022	N	N	Y	N	N	Diabetic rat model	15	N		
Tang, Fengling [55]	2021	Human neonatal foreskin epidermal keratinocytes. Only early passages were used (passages 3–7)	Y	N	N	N	Diabetic rat model	14	N		
Xie, Weinhan [25]	2019	Mouse macrophages (RAW 264.7 cells, ATCC, Manassas, Virginia USA)	Y	N	N	N	Diabetic rat model	14	N		
Zhu, Shuangli [56]	2022	HUVECs	Y	Y	N	Y	Diabetic rat model	14	Y	N	N

Y = yes, N = no; * does not focus on the angiogenic properties of bioactive glass.

**Table 3 ijms-25-01152-t003:** Characteristics and purpose of the hydrogel-based dressings incorporating bioactive glass (BG) of the articles included in this systematic review.

Author	Material	Description	Purpose of the Dressing
Chen et al., 2021 [41]	n Ce-BG/GelMA n = 0, 2, 5	Bioactive glass doped with cerium incorporated into a GelMa (methacrylated gelatin)-based hydrogel. BG composition: 75% SiO_2_, 5% P_2_O_5_, (15-m) CaO mCeO_2_. (m = 0, 2 or 5), molar percentages.	Promotes revascularization and antimicrobial properties.
Hu et al. (2021) [43]	SIS/MBG@Exos	Three-dimensional scaffold dressing composed by decellularized small intestinal submucosa (SIS) containing variety of growth factors, mesoporous bioactive glass (MBG), and exosomes. BG composition: SiO_2_, CaO, and P_2_O_5_ composition is not specified.	The scaffold promotes oxygen and nutrient perfusion, effective drainage of wound secretions, and removal of toxic substances. It also allows for sustained and stable release of exosomes.
Kong et al. (2018) [46]	SA-BG-DFO-GDL	Injectable hydrogel system. Sodium alginate (SA) containing both BG and desferrioxamine (DFO), which is an angiogenesis-promoting drug. Gluconolactone (GDL) is used to promote the release of Ca^2+^ ions from the BG leading to the cross-linking of the hydrogel. BG composition: silica-based, not specified.	Creates a moist environment, necessary for wound healing, and also allows for controlled release of Si^4+^ and Ca^2+^ ions.
Li et al. (2020) [48]	PAB-n and PABC-n	Bioactive self-healing antimicrobial injectable hydrogel scaffold. Main network of polyethylene glycol diacrylate (PEGDA- forming scaffold, with an auxiliary network formed between bioactive glass nanoparticles (BGN) and sodium alginate (SA). PABC refers to BG doped with Cu (BGN/C). The PAB scaffolds with 0 mg/mL, 1 mg/mL, 2 mg/mL, 3 mg/mL BGN were denoted as PAB-0, PAB-2, and PAB-3, respectively. The PABC scaffolds with 3 mg/mL BGNC were denoted as PABC-3. BG composition: Silica based BG nanoparticles doped with Cu, composition not specified.	Mimics extracellular matrix (ECM).
Li et al. (2022) [49]	SA/BG-MMP9-siNP	Injectable hydrogel system composed of sodium alginate (SA) and bioactive glass 45S5 (BG), loaded with MMP9-siNP. siNP: siRNA chitosan nanoparticles; MMP: matrix metalloproteinase; siRNA: silencing RNA. BG composition: 45 wt% SiO_2_, 24.5 wt% CaO, 24.5 wt% Na_2_O, 6.0 wt% P_2_O_5._	Reduces the MMP-9 expression in tissue-forming cells and enhances the synthesis of extracellular matrix proteins.
Ma et al. (2020) [52]	SA/BG-SA (CM)-PLGA(PFD) CM = conditioned medium PFD = Poly(9,9-di-n-dodecylfluorene)	Injectable hydrogel system that sequentially delivers bioactive substances adequate for each stage of wound healing. SA_CM_ corresponds to sodium alginate nanoparticles that enclose bioactive-glass-conditioned medium; these particles are slowly released during the proliferation stage. PFD is released at the end stage and used to regulate ECM synthesis and inhibit angiogenesis, preventing excessive fibrosis and scar tissue and promoting full thickness regeneration. BG composition: 45S.	Covers all of the four stages of wound healing.
Zhu et al. (2022) [56]	HQBn, where n corresponds to the percentage of Magnesium present in the BG, n = 0, 2, 5, 10	Magnesium-containing bioactive glasses (BGs) with a uniform spherical morphology and amorphous structure incorporated into an HQ hydrogel to achieve a nanocomposite hydrogel (HQB). HQ hydrogel consists of an HA-PBA-QCS hydrogel. HA: hyaluronic acid; PBA: phenyl boronate acid; QCS: quaternized chitosan. BG composition: different BG compositions containing Si, Ca, P, and Mg. Compositions reported as weight percentages.	Multifunctional hydrogel with self-healing properties, injectable, adhesive, and antimicrobial performance.
		0MgBG: 83.69% Si, 14.17% Ca, 2.14% P. 2MgBG: 81.54% Si, 12.85% Ca, 2.93% P and 2.67 Mg. 5MgBG: 85.05% Si, 7.67% Ca, 1.71% P and 5.57 Mg 10MgBG: 87.35.% Si, 3.76% Ca, 1.33% P and 7.56 Mg.	

**Table 4 ijms-25-01152-t004:** Evaluation of the angiogenic properties in vitro for different hydrogel-based materials of the articles included in this systematic review.

Material	Assay and Cells Used	Evaluation of Angiogenesis In Vitro
n Ce-BG/GelMA; n = 0, 2, 5 5/G [41]	Tube formation assay using HUVECs	5/G had the most remarkable tube formation effect out of all of the materials containing different Ce concentrations.
SIS/MBG@Exos [43]	Tube formation assay Cells used: HUVECs	No evaluation of the effect of BG, assessment of the effect of exosomes.
SA-BG-DFO-GDL [46]	Tube formation assay VEGF expression Cells used: HUVECs	Tube formation assay: BG+DFO promoted tube formation capacities better than the other groups (BG/SA, DFO/SA, and control). VEGF gene expression in HUVECs cultured with BG+DFO was almost 2 times higher than that in HUVECs cultured with BG and DFO.
PABC [48]	Tube formation assay Cells used: EPCs	EPCs treated for 48 h with the PABC scaffold significantly enhanced their angiogenic ability, evidenced by the higher number of newly formed sprouting tubes in comparison to PA scaffold and control group. PAB scaffold also showed a positive effect on the tube formation of EPCs, indicating that the addition of bioactive glass could significantly enhance the angiogenic ability of PA matrix.
SA/BG-SA (CM)-PLGA(PFD) CM = conditioned medium PFD = Poly(9,9-di-n-dodecylfluorene) [52]	VEGF expression and angiogenic receptors KDR and eNOs Cells used: co-L929 and co-MAEC	CM-BG 1/128 showed a higher stimulatory effect on growth factor VEGF and angiogenic receptors KDR and eNOS than the control.
BG containing Mg [56]	Evaluation of expression levels of CD31, VEGF, KDR, eNOS Cells used: HUVECs	The CD31, VEGF, KDR, eNOS, and angiogenic factors expression of Mg-containing BGs extracts groups was significantly higher than 0MgBG extract group and the blank group, and was upregulated with the increase in Mg concentration, illustrating that Mg^2+^ could regulate the formation of blood vessels. 0MgBG extract group had greater CD31 expression than the blank group, which indicates that BGs also promoted vascularization of endothelial cells.

**Table 5 ijms-25-01152-t005:** In vitro evaluation of the antimicrobial properties for different hydrogel-based materials included in this systematic review.

Material	Bacteria Survival Rate		Survival Rate Time Point	Antimicrobial Assay
	*S. aureus*	*E. coli*		
5/G [41]	(32.81 ± 4.18)%	(9.59 ± 0.43)%	24 h	Agar plate
PABC-3 [48]	~0%	~0%	3 h	Agar plate
HQ [56]	(4.13)%	(7.63)%	Not specified	Spread plate
HQB0 [56]	(2.01)%	(2.83)%		Spread plate
HQB10 [56]	(4.92)%	(1.12)%		Spread plate

**Table 6 ijms-25-01152-t006:** In vivo evaluation of the wound healing capacity and angiogenesis of the different hydrogel-based materials included in this systematic review.

Author (Year)	Animal Model	Wound Size (mm)	Closure Percentage		Wound Follow-Up Period (Days)	Angiogenesis
			New Dressing (%)	Control (%)		
Chen et al. (2021) [41]	Diabetic rat model	10	G: 88.7 0/G: 88.3 5/G: 94.9 *	Gauze: 85.4	21	CD31 and alpha-SMA were significantly higher in the new dressing groups.
Hu et al. (2021) [43]	Diabetic rat model	15	SIS/MBG@Exos: ±95 SIS/@Exos: ±73	Untreated: ±66	14	Collagen I, III, CD31, VEGF, and alpha-SMA were significantly higher, and improved wound perfusion (Doppler) in the SIS/MBG@Exos compared to control.
Kong et al. (2018) [46]	Diabetic rat model	20	SA-BG: 80.3 * SA-DFO: 77 * SA-BG/DFO: 92.6 *^,#^	Untreated: 60.3	20	SA-BG/DFO had a significantly higher expression of VEGF and HIF-1alpha at day 12, followed by a significantly lower expression at day 20. SA-BG/DFO had significantly more (mature) blood vessels at day 12 compared to other groups, followed by significantly fewer new (mature) vessels at day 20.
Li et al. (2020) [48]	Diabetic mouse model	8	PABC: ±88 *^,#^ PAB: ±65 PA: ±61	Untreated: ±58	21	PABC always had the highest blood flow (laser Doppler analysis). There was more mature blood vessels and significantly higher CD31 expression at day 7.
Li et al. (2022) [49]	Diabetic rat model	14	R-siNP: 93 * BG/SA: 94 * R siNP-BG/SA: 98 *	Untreated: 90	12	More neovascularization and mature vessels in new dressings. BG/SA had best angiogenic properties.
Ma et al. (2020) [52]	Diabetic mouse model	10	SA/BG-SA-PLGA: ? SA/BG-SA-PLGA_PFD_: ? SA/BG-SA_CM_-PLGA: ±100 SA/BG-SA_CM_-PLGA_PFD_: ±100	Untreated: ?	18	More neovascularization and mature vessels in new dressing, of which SA/BG-SACM-PLGAPFD showed the highest expression in the early stage of wound healing.
Zhu et al. (2022) [56]	Diabetic rat model	10	HQ: 81.81 * HQB0: 88.58 * HQB10: 98.38 *^,#^	Untreated: 76.54	14	HQB10 had the highest CD31 expression and more mature blood vessels, with a decline at day 14.

* Significant difference compared to control group; # significant difference compared to other new dressings; ? no values provided.

**Table 7 ijms-25-01152-t007:** Characteristics and purpose of the bioactive glass (BG)-based fibers synthesized by the different authors of the articles included in this systematic review.

Author (Year)	Material	Description	Purpose of the Dressing
Chen et al., (2019) [40]	Trilayer scaffold: chitosan/PVA/PVAnBG	Trilayer nanofibrous membrane. Sublayer membrane: electrospun chitosan and then chitosan and polyvinyl alcohol (PVA) were concurrently electrospun on top of chitosan membrane as mid-layer using a two-channel syringe pump. The top layer consisted of PVA blended with nBG. The thickness of each layer was controlled by electrospinning time. BG composition: 80% SiO_2_, 16% CaO, and 4% P_2_O_5_, molar percentages [60].	The chitosan layer has antimicrobial properties, which protects the wound from infection. The middle layer consists of hygroscopic PVA, and it absorbs wound exudate and provides a suitable moist microenvironment for wound proliferation. The top layer PVA/nBG contributes to skin regeneration by the release of bioactive ions (Si and Ca).
Gao et al., (2017) [42]	CPB and CP scaffolds CPB: collagen, PCL (polycaprolactone) and BG CP: collagen and PCL	Bioactive nanofibrous matrix composed of ECM-componential collagen (Col, mimicking protein), polycaprolactone (PCL), and bioactive glass nanoparticles (BGNs, mimicking biological apatite) (CPB). CPB and CP (col/PCL) nanofibrous scaffolds with a diameter of 300–500 nm were prepared using an electrospinning technique. CPB = collagen, polycaprolactone, B = BG. BG composition: 60% SiO_2_, 36% CaO, and 4% P_2_O_5_, molar percentages.	Enhances healing in diabetic full-thickness wounds.
Jana et al., (2022) [44]	F-1 (fish collagen + BG powders)—Fcol/BG microfibers F-2 (fish collagen + Cu-doped BG powders)—Fcol/CuBG microfibers F-3 (fish collagen + Co-doped BG powders)—Fcol/CoBG microfibers F-4 (fish collagen + Cu- and Co-doped BG powders)-Fcol/CuCoBG microfibers	Electrospun fish collagen with BG powders. The electrospun microfibers have a porous architecture, mimicking the structural features of the ECM. The collagen used is from Rohu fish. BG composition: Na_2_O-CaO-SiO_2_-TiO_2_-B_2_O_3_-P_2_O_5_ with Cu (up to 3 wt.%) or Co (up to 5 wt.%) oxide used as dopants. No detailed composition specified.	Accelerates wound healing through stimulation of key events such as angiogenesis and ECM reconstruction under diabetic conditions. ECM structure is obtained through the electrospinning process.
Jiang et al., (2020) [45]	BG/PDA/PM	Micropatterned scaffold with surface-loaded spherical bioactive glasses. PDA: polydopamine, PM: refers to the scaffold without the PDA coating. BG composition: 80% SiO_2_, 15% CaO, 5% P_2_O_5_. Does not specify if percentages are weight or molar.	The micro- to nanoscaled hierarchical structure and PDA coating could mimic the structure and function of the native extracellular matrix (ECM), and the scaffold could sustainably release the Si and Ca ions, enhancing wound healing.
Li et al., (2017) [47]	PEM: patterned electrospun membrane BG/PEM	Patterned electrospun membrane (PEM) consisting of a mixture of polycaprolactone and poly (DL-lactic acid) covered by BG nanocoatings, using pulse electric deposition (BG/PEM). BG composition: Ca_0.25_ P_0.05_ Si_0.7_ O_5.2._	Accelerates diabetic wound healing through the stimulation of angiogenesis.
Sharaf et al. (2022) [54]	BGNP CA BGNP: bioactive glass nanoparticles CA: cellulose acetate	Cellulose acetate electrospun nanofiber composite. BG composition: BG nano powder purchased from Merck (*p* ≥ 98%). Composition not specified.	Promotes rapid and efficient wound healing. Has antimicrobial properties.
Zhang et al. (2020) [53]	Composite scaffold 4 types: Unordered PLA (UP) UP/Gel (UPG) Patterned PLA/Gel (PG)	Micropatterned nanofibrous scaffold with bioglass nanoparticles encapsulated inside coaxial fibers, prepared by electrospinning. The resulting scaffolds are PLA (UP), UP/Gel (UPG), and patterned PLA/Gel (PG), where U stands for unordered and P for BG@PLA/Gel (BG@PG) scaffold. PLA: polylactic acid, Gel: gelatin BG composition: ratio of 5 Si: 1 Ca. Detailed composition not specified.	The structure allows for sustained Si ion release, and it also improves hydrophilicity of the scaffold. Hydrophilicity is more conductive to cell adhesion and growth. The scaffold’s porosity provides space for cell growth and is more conducive to nutrient transport.

**Table 8 ijms-25-01152-t008:** In vivo evaluation of the wound healing capacity and angiogenesis of the different electrospun fibers and scaffolds included in this systematic review.

Author (Year)	Animal Model	Wound Size (mm)	Closure Percentage		Wound Follow-Up Period (Days)	Angiogenesis
			New Dressing (%)	Control (%)		
Chen et al., (2019) [40]	Diabetic mouse model	10	TFM: ? nBG-TMF: ±80	Petrolatum gauze: ±66	21	VEGF, TGF-beta were significantly upregulated in the nBG-TMF group compared to control.
Li et al. (2017) [47]	Diabetic mouse model	13	PEM: 57 BG/PEM: 80	Control: 56	13	Significantly increased CD31, VEGF, collagen I/III expression was found in BG/PEM compared to PEM and control. Neovascularization was significantly higher in BG/PEM at day 15.
Gao et al., (2017) [42]	Diabetic rat model	20	CPB: ±100 *^,#^ CP: ±99	Untreated: 98	21	VEGF expression was significantly upregulated in CPB and significantly more (mature) vessels, which were 23/per mm^2^, were seen in CPB compared to CP and control.
Jana et al., (2022) [44]	Diabetic rabbit model	10	Fcol/BG: 78 * Fcol/CuBG: 78 * Fcol/CoBG: 84 * Fcol/CuCoBag: 88.5 *	Duoderm: 70 Untreated: 27	14 (no values for day 21)	Fcol/CuBG, Fcol/CoBG, and Fcol/CuCoBag have significantly higher blood vessels than other treatments.
Jiang et al., (2020) [45]	Diabetic mouse model	10	P 95.17 PDA/PM: 98.67 BG/PDA/PM: 99.02 *^,#^	Patterned scaffold: 91.87	15	CD31 expression was significantly upregulated in BG/PDA/PM at day 7 and more downregulated at day 15. BG/PDA/PM group always had a higher and more mature blood vessel density.
Zhang et al., (2020) [53]	Diabetic rat model	8	PG: 96.8 BG@PG: 99.3 *^,#^ UPG: 92.7	Untreated: 80.2	14	CD31 expression and VEGF were significantly upregulated in BG@PG compared to all other groups on day 7 and decreased at day 14.
Sharaf et al., (2022) [54]	Diabetic rat model	19	CA-electrospun nanofiber with 3% BGNP: 99.2 *	Unfabricated CA fiber: ?	15	Not assessed.

* Significant difference compared to control group; # significant difference compared to other new dressings; ? no values provided.

**Table 9 ijms-25-01152-t009:** Characteristics and purpose of the miscellaneous bioactive-glass-based materials included in this systematic review.

Author	Material	Description	Purpose of the Material
Bao et al., (2020) [39]	Four-layer composite dressing with a micropore-array-modified Janus membrane	Four-layer composite dressing with a micropore-array-modified Janus membrane for self-pumping and bioactive ion backflow, a superabsorbent layer for high-capacity water absorption, and a bioactive layer containing bioglass for stimulating angiogenesis. It allows for wound exudates transport from wound bed to the dressing, but also enables controlled backflow of bioactive-ion-containing fluid to the wound bed for stimulating angiogenesis. Janus fabric: composed of hydrophilic and hydrophobic film. It can be used to manage humidity of human skin as it has a unidirectional water transport function. BG composition: 45S5: 45% SiO_2_, 24.5% Na_2_O, 24.5% CaO, and 6% P_2_O_5_, weight percentages.	Removes exudate while at the same time allowing bioactive glass ionic products to travel to the wound.
Mao et al., (2014) [51]	Vaseline-based ointment containing 45S5 BG and Yunnan Baiyao (YB) 1. Ointment I: Vaseline+45S5 2. Ointment II: ointment I + 5% YB 3. Ointment III: ointment I + 10% YB 4. Ointment IV: ointment I + 20% YB 5. Ointment V: Vaseline + 16% YB	Mix of bioactive glass powder with Vaseline (P = 84%) and different percentages of YB. BG composition: 45S5: 45% SiO_2_, 24.5% Na_2_O, 24.5% CaO, and 6% P_2_O_5_, weight percentages.	Improves the healing condition of full-thickness diabetic wounds. Yunnan Baiyao is used for hemostatic and anti-inflammatory functions. BG is used to promote vascularization and growth factor production.
Lin et al., (2012) [50]	1.45S5 bioactive glass (45S5) 2. 58S sol–gel bioactive glass (SGBG-58S) 3. Nanobioactive glass (NBG-58S)	Both SGBG-58S and NBG-58S have a porous microstructure, while 45S5 has an impacted microstructure; thus, the bioactivity of SGBG-58S and NBG-58S is higher than that of 45S5. BG composition:	BG could be easily squeezed from the tube and daubed onto the desired surface.
		45S5: 45% SiO_2_, 24.5% Na_2_O, 24.5% CaO, and 6% P_2_O_5_, weight percentages. 58S: 58% SiO_2_, 33% CaO and 9% P_2_O_5_, weight percentages.	
Tang et al., (2021) [55]	58S-BG ionic extracts Dilution ratios: 1/20, 1/80, 1/320	Ionic extract (BG extract is injected subcutaneously). Ionic extracts were prepared by adding 1g of sterilized BG powder into a 10 cm culture dish and soaking it in 5 mL of Dermal Cell Basal Medium. After 24 h, incubation samples were centrifuged and filtrated to obtain the extracts. For cell culture, BG extracts were diluted at ratios 1/20, 1/80, and 1/320. BG composition: 60% SiO_2_, 36% CaO, and 4% P_2_O_5_, molar percentages.	Accelerates re-epithelization and stimulates keratinocyte differentiation.
Xie et al., (2019) [25]	Vaseline-based ointment containing BG	Spherical particles of BG with an average size of 551 ± 137.8 nm incorporated into Vaseline at different concentrations. BG composition: 60% SiO_2_, 36%CaO, and 4% P_2_O_5_, molar percentages.	Promotes macrophages to secrete anti-inflammatory factors and switch to the M2 phenotype, aiding wound healing.
	HBG (high BG concentration) LBG (low BG concentration)		

**Table 10 ijms-25-01152-t010:** In vivo evaluation of the wound healing capacity and angiogenesis of the miscellaneous treatments included in this systematic review.

Author (Year)	Animal Model	Wound Size (mm)	Closure Percentage		Wound Follow-Up Period (Days)	Angiogenesis
			New Dressing (%)	Control (%)		
Bao et al., (2020) [39]	Diabetic mouse model	13	20BG: ±100 *^,#^ 0BG: ±97 *	Tegaderm: 93	13	20BG had a significantly higher blood vessel density and CD31 expression in comparison to control and 0BG. 0BG had significantly better angiogenic properties than control.
Mao et al., (2014) [51]	Diabetic rat model	30	Ointment I: ±98 Ointment II: ±98^#^ Ointment III: ±97 Ointment IV: ±98 Ointment V: ±96	Saline: ±84 Vaseline: ±93	21	Immunofluorescence staining showed that ointment II promoted more blood vessel formation than the Vaseline-treated group.
Lin et al., (2012) [50]	Diabetic rat model	18	45S5: ±99 SGBG-58S: 100 * NBG-58S: ±99 *	Vaseline: ±93	14	Positive VEGF staining was observed in all tissue samples. At day 7, the regenerated tissues of the BG groups showed more VEGF than control. SGBG-58S and NBG groups had more blood vessels than 45S5 group. The expression of VEGF at day 14 was lower than at day 7.
Tang et al., (2021) [55]	Diabetic rat model	18	1/20 BG: 86.29 *^,#^	Blank: 69.20 Control: 73.23	14	Not assessed.
Xie et al., (2019) [25]	Diabetic rat model	18	HBG: ±75 LBG: ±98	Vaseline: ±88	14	Not assessed.

* Significant difference compared to control group; # significant difference compared to other new dressings.

**Table 11 ijms-25-01152-t011:** Antibacterial properties of the different bioactive glass compositions used in the materials evaluated in this systematic review.

First Author	Year	Bioactive Glass Composition Used in Dressing	Demonstrated Antimicrobial Properties of the BG Composition
Bao, Feng [39]	2020	45S5 (45% SiO_2_, 24.5% Na_2_O, 24.5% CaO, and 6% P_2_O_5_, weight percentages)	Previous studies have shown that BG 45S5 has antibacterial properties [67]. However, this was not evaluated by the authors.
Chen, Qingchang [40]	2019	80% SiO_2_, 16% CaO, and 4% P_2_O_5_	Previous antimicrobial studies on this exact composition were not found. Authors attributed the material’s antibacterial properties to chitosan and not to their specific bioactive glass composition.
Chen, Yue-Hua [41]	2021	75% mol SiO_2_, 5% mol P_2_O_5_, (15 m) CaO mCeO_2_ (m = 0, 2 or 5) mol	The authors showed that their BG composition reduced colony numbers of both Gram-positive and Gram-negative bacteria.
Gao, Wendong [42]	2017	58S (60% mol SiO_2_, 36%mol CaO, and 4%mol P_2_O_5_)	Antibacterial properties of 58S BG have been found by previous studies [68,69] but were not evaluated by Gao et al. (2017) [42].
Hu, Yiqiang [43]	2021	Silica-based, composition not specified	Antimicrobial properties were not evaluated and exact composition is not known in order to look into previous studies.
Jana, Sonali [44]	2022	Na_2_O-CaO-SiO_2_-TiO_2_-B_2_O_3_-P_2_O_5_ with Cu (up to 3wt%) or Co (up to 5 wt%) oxide used as dopants	Antimicrobial properties were not evaluated by the authors and exact composition is not known in order to look into previous studies.
Jiang, Yuqi [45]	2020	80% SiO_2_, 15% CaO, 5% P_2_O_5_	Antimicrobial properties were not evaluated by the authors and previous evidence of antibacterial properties was not found.
Kong, Lingzhi [46]	2018	Silica-based, composition not specified	Antimicrobial properties were not evaluated by the authors and exact composition is not known in order to look into previous studies.
Li, Jinyan [47]	2017	Chemical formula: Ca(0.25)P(0.05)Si(0.7)O(5.2)	Antimicrobial properties were not evaluated by the authors and previous studies on the exact composition were not found.
Li, Yannan [48]	2020	Silica-based BG nanoparticles doped with Cu, composition not specified	Antibacterial properties were evaluated by the authors and it was concluded that their PABC-3 (3% Cu) scaffold had excellent antimicrobial activity against *S. aureus* and *E. coli*.
Li, Ying [49]	2022	45S5	Previous studies have shown that BG 45S5 has antibacterial properties [67]. However, this was not evaluated by the authors.
Lin, Cai [50]	2012	45S5 58S	Previous studies have shown that BG 45S5 [67] and 58S have antibacterial properties [68,69]. However, this was not evaluated by the authors.
Mao, Cong [51]	2014	45S5	Previous studies have shown that BG 45S5 has antibacterial properties [67]. However, this was not evaluated by the authors.
Ma, Zhijie [52]	2020	45S	Previous studies have shown that BG 45S5 (also known as 45S) has antibacterial properties [67]. However, this was not evaluated by the authors.
Zhang, Pengju [53]	2020	5:1 Si:Ca ratio	Antimicrobial properties were not evaluated by the authors and exact composition is not known in order to look into previous studies.
Sharaf, Samar [54]	2022	BG nano powder purchased from Merck (P ≥ 98%). Composition not specified	Authors showed that the addition of their BG composition increased zone of inhibition in comparison to control for several Gram-positive and Gram-negative bacterial strains.
Tang, Fengling [55]	2021	58S	Antibacterial properties of the 58S BG have been shown by previous studies [68,69] but were not evaluated by Tang et al. (2021) [55]..
Xie, Weinhan [25]	2019	58 S	Antibacterial properties of the 58S BG have been shown by previous studies [68,69] but were not evaluated by Xie et al. (2019) [25]..
Zhu, Shuangli [56]	2022	Different BG compositions containing Si, Ca, P, and Mg. Compositions reported as weight percentages 0MgBG: 83.69% Si, 14.17% Ca, 2.14% P 2MgBG: 81.54% Si, 12.85% Ca, 2.93% P and 2.67 Mg 5MgBG: 85.05% Si, 7.67% Ca, 1.71% P and 5.57 Mg 10MgBG: 87.35.% Si, 3.76% Ca, 1.33% P and 7.56 Mg	Antibacterial properties were studied. However, the authors did not attribute them to the bioactive glass but to the presence of quaternized chitosan.

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
