# Peer review of "Bioactive-Glass-Based Materials with Possible Application in Diabetic Wound Healing: A Systematic Review"

_ijms, 2024, doi:10.3390/ijms25021152_

Round 1

Reviewer 1 Report

Comments and Suggestions for Authors

t The link of overall text is missing

3The introduction section should be written better to highlight the study

4The last paragraph of introduction should be written properly to identify the objective of the paper.

5What are some of the key complications associated with diabetic foot ulcers (DFUs), hindering their healing process?

6How do bioactive glass-based materials contribute to accelerating diabetic wound healing, as highlighted in the systematic review?

7Explanation of figure and table is missing. The figure and tables is not defined properly.

8Check the reference style.

9What specific areas of further research are essential before advancing bioactive glass-based materials to clinical trials in diabetic wound healing?

1Review's organizational structure is incorrect. To make the necessary corrections, look at the most recent literature review.

1 how might the development of a BG-based wound healing material with antimicrobial properties contribute to addressing challenges in diabetic soft tissue wound healing and the fight against antimicrobial resistance (AMR)?

Comments on the Quality of English Language

Moderate changes to the level of English would be beneficial, so editing of the draft by someone with a good command of English would be quite beneficial.  

Author Response

Thank you for your comments and suggestions, up next you will find the reply to your comments and the changes we have made to the manuscript to include them.

  1. Comment: Review's organizational structure is incorrect. To make the necessary corrections, look at the most recent literature review.

Reply: We followed the review structure indicated by the PRISMA guidelines. The structure suggested by the guidelines is: title, abstract, introduction, methods, results, discussion, others.

  1. Comment: The overall link is missing.

Reply: It is not entirely clear to us which specific link is missing. We would appreciate if the comment could be more detailed in order to be able to address it properly.

  1. Comment: The introduction section should be written better to highlight the study

Reply: Points 1.1 and 1.2 from the introduction were merged in order to make the objective clearer. We also added a sentence to the abstract to emphasize the objective of the review.

  1. Comment: The last paragraph of introduction should be written properly to identify the objective of the paper.

Reply: Page 4, line 120. The objective of the paper was written down in more detail.

“Given the increasing interest in utilizing bioactive glass in soft tissue regeneration, this review aims to create an overview of the composition and purpose of the novel bioactive glass-based materials developed for the treatment of diabetic soft tissue wounds. In addition the cytocompatibility, angiogenic and antimicrobial properties are evaluated in-vitro and in-vivo. Diabetic wound closure and wound healing quality of the different bioactive glass based materials is evaluated in-vivo. This systematic review intends to contribute to the advancement of diabetic wound healing therapeutic strategies development.”

  1. Comment: What are some of the key complications associated with diabetic foot ulcers (DFUs), hindering their healing

Reply: Page 2, line 44 “These ulcers form due to a combination of factors, such as lack of feeling in the foot, poor circulation, foot deformities, irritation (such as friction or pressure), and trauma, as well as duration of diabetes. DFUs affect around 15% of diabetic patients, and if left untreated, they can lead to lower extremity amputation (14 - 24%), with a mortality rate of 50-59% within five years post-amputation. Patients with DFU can experience discomfort, swelling, drainage, skin changes in the surrounding areas , the foot ulcer can also cause to loss of mobility, which can lead to depression , they. The impaired healing of diabetic ulcers can be attributed to a range of factors, including pre-existing conditions such as peripheral neuropathy, arterial disease, immunodeficiency, infection, and aberrant extracellular matrix (ECM) degradation.”

  1. Comment: How do bioactive glass-based materials contribute to accelerating diabetic wound healing, as highlighted in the systematic review?

Reply: Page 4, line 107. “For soft tissue it has been shown that specific bioactive glass compositions can facilitate wound healing by promoting angiogenesis, having antibacterial properties and their ability to enhance the M1 to M2 macrophage switch , accelerating the wound healing process.”

  1. Comment: Explanation of figure and table is missing. The figure and tables is not defined properly.

Reply: Figure 3 was removed as it did not add much additional information. A better description was added to figure 2. The table titles were also modified to promote a clearer understanding of the tables.

  1. Comment: Check the reference style

Reply: The reference style is Vancouver, a reference manager (EndNote) was used for the citations. If the reference style needs to be changed to a specific format please let us know and we will gladly do so.

  1. Comment: What specific areas of further research are essential before advancing bioactive glass-based materials to clinical trials in diabetic wound healing?

Reply: Page 30, line 859. Before advancing BG-based materials to clinical trials, research should focus on in-vivo diabetic animal models and study the difference between animals treated with the BG based material, no treatment and a control, to properly identify the effect of the new material in the wound healing process. Research should also focus on the antimicrobial properties of these materials, since as mentioned previously could help in the fight against AMR.”

  1. Comment: How might the development of a BG-based wound healing material with antimicrobial properties contribute to addressing challenges in diabetic soft tissue wound healing and the fight against antimicrobial resistance (AMR)?

Reply: Page 28, line 806. “Bioactive glass based materials can be a valuable tool in the fight against antimicrobial resistance (AMR) since so far there has not been shown evidence that bacteria can create resistance to the antibacterial mechanism of bioactive glass. However, further studies are needed to deepen the knowledge of specific antibacterial and antbiofilm activity and improve the use of bioactive glass in the fight against AMR.”

The changes can be seen highlighted in yellow in the manuscript.

Reviewer 2 Report

Comments and Suggestions for Authors

The manuscript seems to describe how to use the PRISMA flow diagram in healing wounds of diabetic foot issues (especially type I which is a minority) using bioglass-containing materials.

Comments:

Page 4- line 109- a further description of the “more open silicate network should be included.

Page 4- PriSMA should be used in extenso prior abbreviation.

Page 6- According to the PRISMA flow, only 19 papers within the 2017-2022 range were included in the review. The compositions of a few bioglasses were not known. Are they enough to obtain the current review? It seems a rather simplified approach.

Page 29, line 807- Only the composition factor is discussed while the textural properties were not mentioned.

Author Response

Thank you for your comments and suggestions, up next you will find the reply to your comments and the changes we have made to the manuscript to include them.

  1. Comment: The manuscript seems to describe how to use the PRISMA flow diagram in healing wounds of diabetic foot issues (especially type I which is a minority) using bioglass-containing materials.

Reply: Thank you for your comment. We just want to clarify that the PRISMA guidelines and flow diagram are the tools that we used to create our systematic review. We assess different bioactive glass-based materials and their potential to be used in the treatment of diabetic wounds.

  1. Page 4, line 109. The following text was added in order to better describe why the silicate networks opened. “ The presence of these network modifiers opens the silicate network by breaking Si-O-Si, which causes the pronounced differences in properties of bioactive glass and conventional glass”
  2. Page 4, line 122. The extenso version of PRIMSA was added, Preferred Reporting Items for Systematic Reviews and Meta-Analyses
  3. Comment: Page 6- According to the PRISMA flow, only 19 papers within the 2017-2022 range were included in the review. The compositions of a few bioglasses were not known. Are they enough to obtain the current review? It seems a rather simplified approach.

Reply: Considering the research question and the scope of the review the amount of papers should be sufficient. Of course a larger amount of papers would permit a more comprehensive review, however there is no more literature available for our specific research question: bioactive glass based materials for diabetic wound healing that matched our selection criteria. It is also relevant to mention that we registered a protocol for our systematic review ,before starting the screening process, using the Open Science Framework (OSF), and once the protocol is registered no further changes can be made in order to include more articles in the review (Registration DOI: https://doi.org/10.17605/OSF.IO/BFHPM )

  1. Page 29, line 792. The relevance of texture for antibacterial properties was added with the following paragraph: “It is relevant to mention that the antibacterial properties of BG are determined by its composition, size and concentration. These properties are attributed to an increase in pH and osmotic pressure. Particle size and texture are also relevant since high specific surface area increases the ability to efficiently release therapeutic ions as well as increase pH and osmolality faster.”

The changes can be seen highlighted in yellow in the manuscript.

Reviewer 3 Report

Comments and Suggestions for Authors

This review is a thorough and detailed description and summarization of the current state of bioactive glasses in diabetic wound healing. This is a very interesting and pressing issue since the innovations in wound treatment are urging and important.

This manuscript definitely a good summarization of the state-of-the art regarding the given topic. In my opinion, it provides a significant contribution to the scientific literature.

In addition, this review also contains new and novel information that is useful to the readers.

The methodology of the paper is clearly described and easy to understand and follow.

The Figures and Tables provided are all clear, informative, and correct.

The conclusion is consistent with the topic addressed, however, too concise and does not sufficiently highlight the final evidence and challenges of the application of BGs in wound healing. I would advise expanding the conclusions by going into more detail and emphasizing the most relevant achievements and difficulties.

Author Response

Thank you for your comments and suggestions, up next you will find the reply to your comments and the changes we have made to the manuscript to include them.

  1. We really appreciate your positive comments, but believe that the star grade given at the beginning in the questions for general evaluation does not match them, as we were granted a 1 star grade for all 5 items and comments seem quite positive for this to be the case. 
  2. Comment: The conclusion is consistent with the topic addressed, however, too concise and does not sufficiently highlight the final evidence and challenges of the application of BGs in wound healing. I would advise expanding the conclusions by going into more detail and emphasizing the most relevant achievements and difficulties.

Reply: The conclusion was extended according to your suggestion. Page 31, line 873.

“Bioactive glass (BG) based materials, as discussed in this systematic review, indicate potential benefits by accelerating all of the stages involved in diabetic wound healing and enhancing overall wound healing quality. Wound healing improves due to the release of therapeutic ions from the bioactive glass based materials, increasing angiogenesis, promoting M1 to M2 phenotype macrophage switching and in most cases avoiding infection due to antibacterial properties. A comparison of the effectiveness of the different materials is not possible, since the models used by the studies were not equivalent and the outcome measurements are not standardized.

It is essential to note that further research on determining the optimal BG concentration to promote angiogenesis, M1 to M2 macrophage phenotype switching and wound healing in general, identifying the ideal dressing and most importantly, understanding the antimicrobial properties of BG based materials are warranted to eventually advance to clinical trials. In the near future and with more research, a BG based wound healing material with antimicrobial properties could address the challenges in diabetic soft tissue wound healing and hence greatly assist in the fight against AMR and improve the treatment of diabetic wound infections.’’

The changes can be seen highlighted in yellow in the manuscript.